# Optical clocks at sea

Jonathan D. Roslund[1✉], Arman Cingöz[1], William D. Lunden[1], Guthrie B. Partridge[1], Abijith S. Kowligy[1], Frank Roller[1], Daniel B. Sheredy[1], Gunnar E. Skulason[1], Joe P. Song[1], Jamil R. Abo-Shaeer[1] & Martin M. Boyd[1✉]

Deployed optical clocks will improve positioning for navigational autonomy[1], provide remote time standards for geophysical monitoring[2] and distributed coherent sensing[3], allow time synchronization of remote quantum networks[4,5] and provide operational redundancy for national time standards. Although laboratory optical clocks now reach fractional inaccuracies below $10^{-18}$ (refs. 6,7), transportable versions of these high-performing clocks[8,9] have limited utility because of their size, environmental sensitivity and cost[10]. Here we report the development of optical clocks with the requisite combination of size, performance and environmental insensitivity for operation on mobile platforms. The 35 l clock combines a molecular iodine spectrometer, fibre frequency comb and control electronics. Three of these clocks operated continuously aboard a naval ship in the Pacific Ocean for 20 days while accruing timing errors below 300 ps per day. The clocks have comparable performance to active hydrogen masers in one-tenth the volume. Operating high-performance clocks at sea has been historically challenging and continues to be critical for navigation. This demonstration marks a significant technological advancement that heralds the arrival of future optical timekeeping networks.

Atomic timekeeping plays an essential role in modern infrastructure, from transportation to telecommunications to cloud computing. Billions of devices rely on the Global Navigation Satellite System for accurate positioning and synchronization[11]. The Global Navigation Satellite System is a network of distributed, high-performance microwave-based atomic clocks that provide nanosecond-level synchronization globally. The emergence of fieldable optical timekeeping, which offers femtosecond timing jitter at short timescales and multiday, subnanosecond holdover, along with long-distance femtosecond-level optical time transfer[12], paves the way for global synchronization at picosecond levels.

Molecular iodine ($I_2$) has a legacy as an optical frequency standard[13–17]. Several iodine transitions are officially recognized as length standards[18], and the species underpinned one of the first demonstrations of optical clocks[19,20]. More recently, iodine frequency standards have been investigated for space missions[21–24]. Here we report the deployment of several high-performance, fully integrated iodine optical clocks and highlight their ability to maintain nanosecond (ns)-level timing for several days while continuously operating at sea.

These clocks use a robust vapour cell architecture that uses no consumables, does not require laser cooling or a prestabilization cavity and is first-order insensitive to platform motion. Similar approaches with rubidium vapour cells are under development[25–27]. Importantly, iodine clocks use mature laser components at 1,064 nm and 1,550 nm. The focus on a robust laser system rather than a high-performance atomic species resolves system-level issues with dynamics, lifetime, autonomy and cost. Although not as accurate as laboratory optical clocks using trapped atoms or ions, iodine clocks can provide maser-level performance in a compact, robust and mobile package.

Initial clock prototypes were integrated into 35 l, 3 U 19-inch rackmount chassis, shown in Fig. 1a. Clock outputs are at 100 MHz, 10 MHz and 1 pulse per second. Auxiliary optical outputs are provided for the frequency comb and clock laser (1,550 nm and 1,064 nm, respectively). The physics packages, which include the spectrometer, laser system and frequency comb, were designed and built in-house to reduce system-level size, weight and power (SWaP). Field-programmable gate array-based controllers perform digital locks for the laser and frequency comb, servo residual amplitude modulation (RAM) and stabilize the pump and probe powers. The clock operates using a commercial 1 U rackmount power supply and control laptop. Each system consumes about 85 W (excluding the external power supply) and weighs 26 kg.

Two clocks with identical hardware (PICKLES and EPIC) were developed with physics packages targeting short-term instability below $10^{-13}/\sqrt{\tau}$, comparable to commercial masers. A third clock (VIPER) with a relaxed performance goal of less than $5 \times 10^{-13}/\sqrt{\tau}$ was built using a smaller iodine spectrometer and simplified laser system to reduce the physics package volume by 50% and power consumption by 5 W; the chassis volume was unchanged. The frequency comb design and control electronics for PICKLES, EPIC and VIPER are largely identical.

In April 2022, PICKLES and EPIC were shipped to the National Institute of Standards and Technology (NIST) in Boulder, Colorado for assessment against the Coordinated Universal Timescale maintained at NIST, that is, UTC(NIST)[28]. The clocks operated on an optical table without any further measures to insulate them from the NIST laboratory environment, which is temperature stabilized. The laboratory was also in active use throughout the measurement campaign. The 10 MHz tone from each clock was compared against a 5 MHz maser signal with a Microchip 53100A phase noise analyser in a three-cornered hat (TCH) configuration. NIST maser ST05 (Symmetricom MHM-2010) was selected as the lowest drift maser in the ensemble ($3 \times 10^{-17}$ per day). The measurement scheme allows for decorrelating the three

[1]Vector Atomic, Inc., Pleasanton, CA, USA. ✉e-mail: jon@vectoratomic.com; marty@vectoratomic.com

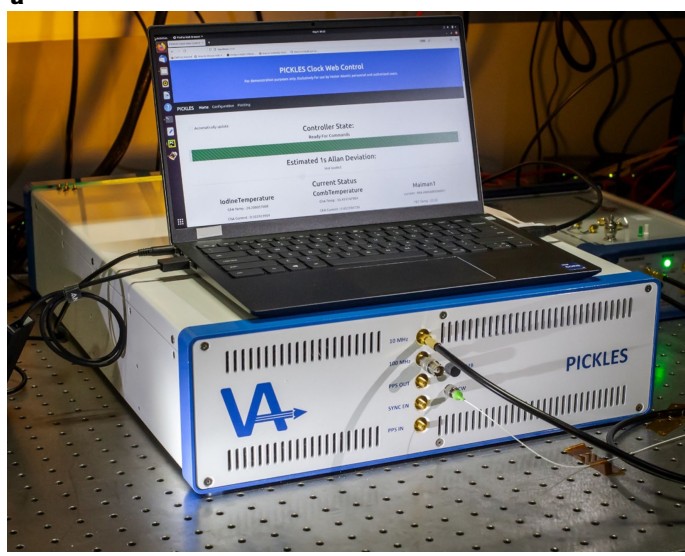

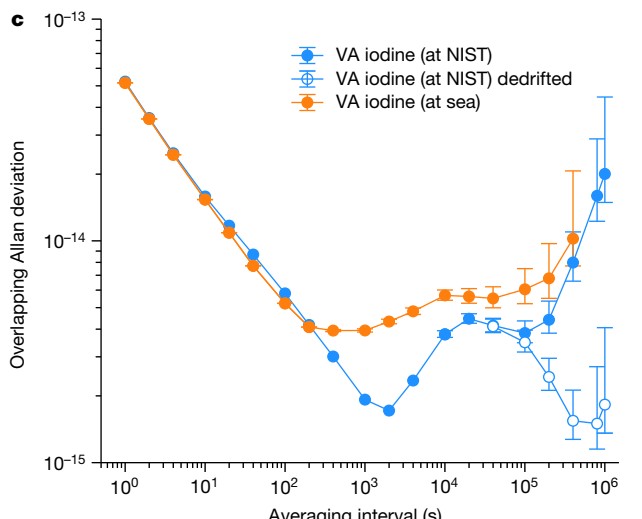

**Fig. 1 | Single-clock performance at NIST and at sea. a**, The 3 U, 19-inch rackmount iodine optical clock occupies a volume of 35 l and consumes less than 100 W. **b**, Measured phase noise for the iodine clock at 10 MHz, 100 MHz and 1,064 nm. **c**, Overlapping Allan deviation for the iodine clock operating at NIST and at sea. At short timescales, the instability in a dynamic environment is identical to the laboratory. The iodine clock can maintain less than $10^{-14}$

frequency instability for several days despite several-degree temperature swings, significant changes in relative humidity and changing magnetic fields. **d**, The clocks can maintain holdovers of 10 ps for several hours and 1 ns for several days, showing their potential as the basis for a picosecond-level timing network.

clocks at short timescales and measuring against the NIST composite timescale AT1, derived from the maser ensemble, at longer timescales. Importantly, ST05 was operated in an environmental chamber in a separate laboratory, providing an environmentally uncorrelated reference. The 1,064 nm optical beatnote between PICKLES and EPIC was simultaneously monitored for cross-validation. After installation, the clocks were left to operate autonomously. The measurement setup was remotely monitored without intervention from our California headquarters, and the comparison was intentionally terminated after 34 days on return to NIST.

The overlapping Allan deviation for the entire 34-day dataset without any windowing, dedrifting or filtering is shown in Fig. 2. To present the individual clock performance, the Allan deviation plot uses the 1–1,000 s instability extracted from TCH analysis and the direct instability against ST05 for time periods longer than 1,000 s (Extended Data Fig. 3). The PICKLES and EPIC short-term instabilities of $5 \times 10^{-14}/\sqrt{\tau}$ and $6 \times 10^{-14}/\sqrt{\tau}$, respectively, outperform the short-term performance of the ST05 maser. Both iodine clocks exhibit fractional frequency

instabilities less than $5 \times 10^{-15}$ after 100,000 s of averaging, equivalent to a temporal holdover below 300 ps after 1 day.

The data also provided an initial measure of the long-term stability of the iodine clocks (Fig. 2, inset). Measured against UTC(NIST), the drift rates for PICKLES and EPIC are $2 \times 10^{-15}$ and $4 \times 10^{-15}$ per day, respectively, consistent with the long-term accuracy of an iodine vapour cell measured over the course of a year[19]. This drift rate is about ten times lower than a typical space-qualified rubidium atomic frequency standard after more than a year of continuous operation[29,30]. Moreover, the iodine-stabilized laser provides a drift rate roughly 10,000–100,000× lower as compared to typical ultralow expansion (ULE) optical cavities[31,32]. This drift rate has been consistent for multiple measurement campaigns over several months (Extended Data Fig. 5). Removal of linear drift from the frequency data indicates that the two clocks continue to hold less than $3 \times 10^{-15}$ instability after more than $10^6$ s (approximately 12 days) of averaging, equivalent to 1 ns timing error over this period. Without drift removal, the long-term clock performance is competitive with the NIST active hydrogen masers; drift removal puts

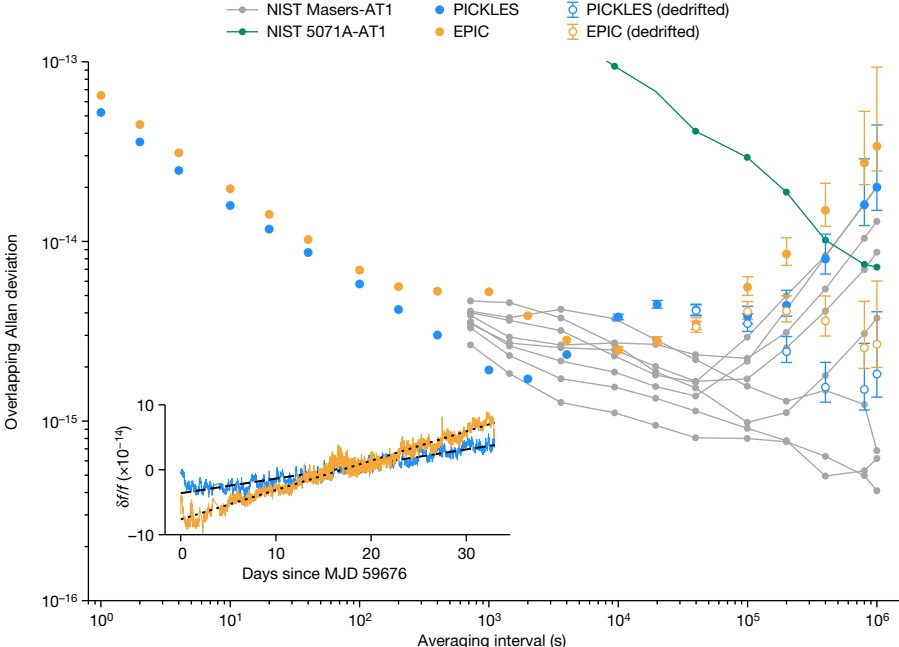

**Fig. 2 | Long-term clock performance.** Overlapping Allan deviation for the 10 MHz outputs of the two iodine clocks measured against the UTC(NIST) timebase for 34 days (blue and orange traces). The clocks exhibit a raw frequency instability of $4 \times 10^{-15}$ (PICKLES) and $6 \times 10^{-15}$ (EPIC) after $10^5$ s of averaging and maintain instability less than $10^{-14}$ for nearly 6 days (PICKLES). With linear drift removal, the frequency instability improves to less than $2 \times 10^{-15}$ (PICKLES) and less than $3 \times 10^{-15}$ (EPIC) for $10^6$ s (open circles). The performance of a variety of NIST masers against the composite AT1 timescale is shown for comparison (grey traces) as well as a commercial caesium clock (green trace). The long-term frequency record for the two iodine clocks against ST05 is shown as an inset. Each trace is shown as a 1,000 s moving average. The linear drift for each clock is observed to be several $10^{-15}$ per day. MJD is the modified Julian day.

the clock instability on par with the highest-performing masers in the NIST bank. Notably, to achieve the drift rates observed in Fig. 2, the NIST masers are operated continuously for years and housed in environmental chambers with a volume of nearly 1,000 l to stabilize temperature and humidity to better than 100 mK and 1%, respectively (ref. 33 and J. Sherman, private communication). The laboratory housing PICKLES and EPIC was stable to hundreds of millikelvins throughout the measurement campaign, which started a few days after a cross-country shipment. Finally, the raw iodine clock performance is below NIST's commercial caesium beam clock (Microchip 5071A) for 5.5 days; the dedrifted iodine performance is below caesium for all observed timescales.

A broad feature with a peak deviation of $4 \times 10^{-15}$ is evident in the PICKLES Allan deviation at roughly 20,000 s (about 7 h) timescales. The equivalent optical frequency deviation of 2 Hz corresponds to a shift of about 2 ppm of the hyperfine transition line centre. We suspect that the origin of this plateau in PICKLES is RAM coupling through a spurious etalon in the spectrometer. By modifying the build procedure, this etalon was mitigated during the build of the EPIC spectrometer.

The iodine clock exhibits excellent phase noise for the 10 and 100 MHz tones derived by optical frequency division as well as the 1,064 nm optical output (Fig. 1b). The phase noise at microwave frequencies is lower than commercial atomic-disciplined oscillators, highlighting the benefits of optical frequency division where the fractional noise of the iodine-stabilized laser is transferred to the frequency comb repetition rate.

Following the measurement against an absolute reference at NIST-Boulder, three optical clocks were brought to Pearl Harbor, HI in July 2022 to participate in the Alternative Position, Navigation and Time (A-PNT) Challenge at Rim of the Pacific (RIMPAC) 2022, the world's largest international maritime exercise. A-PNT was an international demonstration of quantum technologies with academic, government and industry participants. Several prototype quantum technologies

including optical clocks[34,35] and atomic inertial sensors were fielded[36]. The iodine clocks were installed in an open server rack along with a commercial 1 U power supply for each clock, three control laptops and an uninterruptable power supply backup for the system (Fig. 3a). The rack also contained three frequency counters to collect the three pairwise beatnotes and a 53100A phase noise analyser to compare the 100 MHz tone derived from each clock's frequency comb against the other two in a TCH configuration. The total stackup, including three independent clocks, power supplies, computer controls and metrology systems, occupied a rack height of 23 U. The server rack was hard-mounted to the floor of a Conex cargo container, which was craned onto the deck of the New Zealand naval ship HMNZS Aotearoa (Fig. 3b), where it remained during the three weeks the vessel was at sea. Once the ship left port, the three clocks operated without user intervention for the duration of the exercise, apart from one restart of VIPER due to a software fault in the external power supply.

The operating environment during the ship's underway differed significantly from NIST, but the clocks still operated continuously with high performance (Fig. 1c,d). Although the Conex was air-conditioned, the internal environment underwent swings of roughly 2–3 °C peak-to-peak temperature and 4%–5% relative humidity over a day–night cycle. The clock rack was located directly in front of the air conditioning unit, which cycled on and off throughout the day. The clocks also operated continuously through ship motion. The rotational dynamics of the ship included a peak pitch of $\pm 1.5°$ at a rate of $\pm 1.2° \, s^{-1}$ and a peak roll of $\pm 6°$ at a rate of $\pm 3° \, s^{-1}$. Similarly, the maximum surge, sway and heave accelerations were $\pm 0.4$, $\pm 1.5$ and $\pm 1.2 \, m \, s^{-2}$, respectively. A vertical root mean square vibration of $0.03 \, m \, s^{-2}$ (integrated from 1 to 100 Hz) was also experienced. Operation in dynamic environments highlights the robust, high-bandwidth clock readout (greater than 10 kHz control bandwidth) enabled by a vapour cell.

The vessel travelled in all four cardinal directions during the exercise, illustrated by the GPS-tracked trajectory in Fig. 3c. The National Oceanic

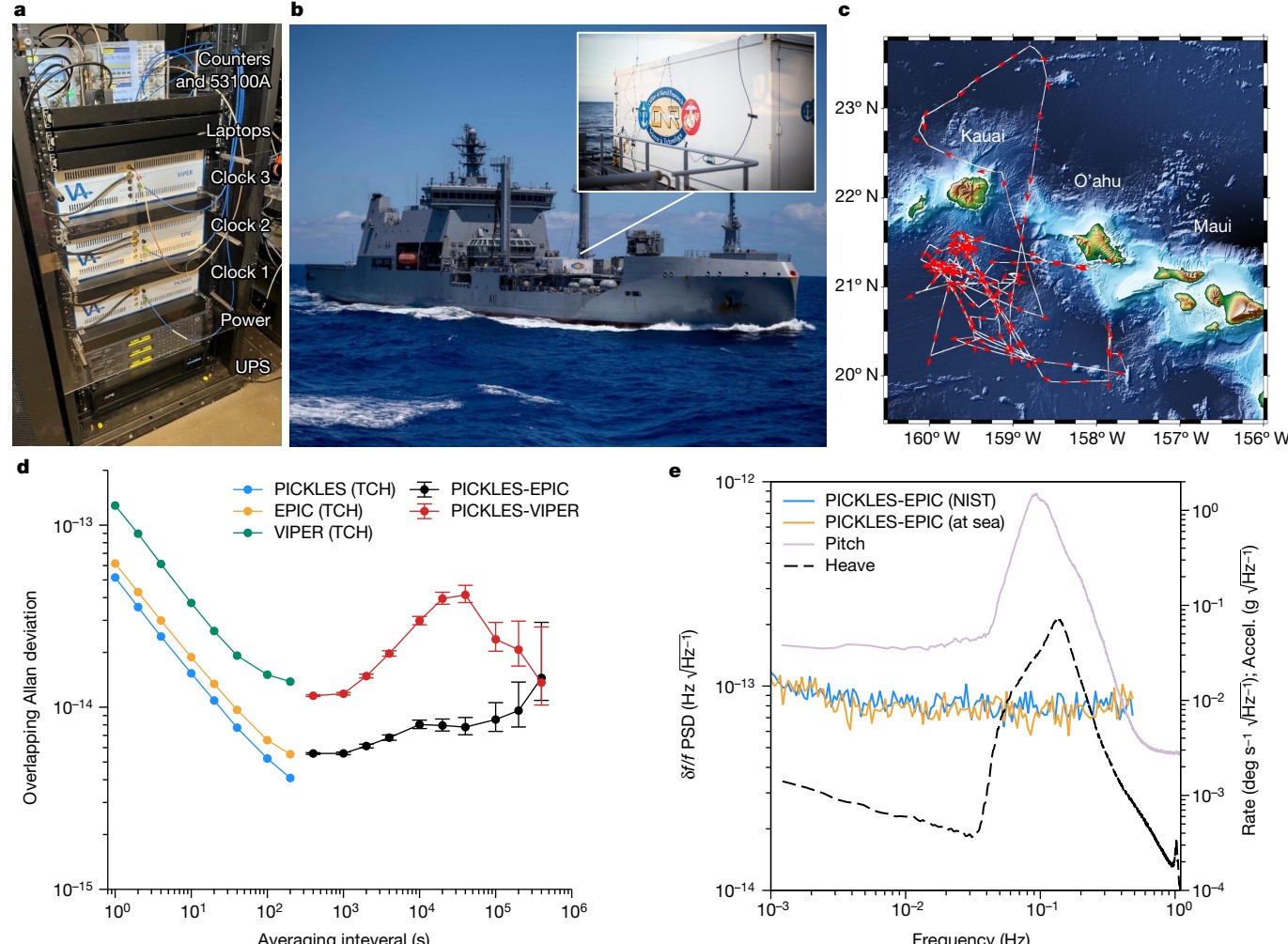

**Fig. 3 | At-sea demonstration of optical clocks. a**, Clock stackup for RIMPAC 2022. The server rack contained three independent optical clocks, a 1 U power supply and control laptop for each clock, an uninterruptable power supply and the measurement system in a total rack volume of 23 U. **b**, The cargo container housing the clocks was craned onto the deck of the HMNZS Aotearoa, where it remained for the three-week naval exercise. **c**, A GPS track of the Aotearoa's voyage around the Hawaiian Islands. The ship started and ended its voyage at Pearl Harbor, O'ahu. **d**, Overlapping Allan deviation during the underway. For time periods less than 100 s, individual clock contributions

are extracted with a TCH analysis; directly measured pairwise instabilities are shown for periods longer than 100 s. The EPIC–PICKLES pair maintains a fractional frequency instability of $8 \times 10^{-15}$ after $10^5$ s of averaging, corresponding to a temporal holdover of 400 ps. **e**, PSD for the PICKLES–EPIC frequency fluctuations at NIST and at sea with the recorded ship pitch and heave (rotation and acceleration on the other ship axes showed similar behaviour). The PICKLES–VIPER PSD (not shown) showed a similar immunity to the ship motion. Photograph of the ship by T. Bacon, DVIDS.

and Atmospheric Administration geomagnetic model for Earth's magnetic field at this latitude and longitude shows that the projection of the Earth's field on the clocks varied by ±270 mG throughout the underway (https://www.ngdc.noaa.gov/geomag/geomag.shtml).

The overlapping Allan deviations measured during the voyage are shown in Fig. 3d. For time periods less than 100 s, the individual clock contributions are extracted with a TCH analysis. Directly measured pairwise instabilities are shown for longer time periods. There was no degradation in the clock signal-to-noise ratio (SNR) despite ship vibration and motion; the short-term performance for the three clocks was identical to that observed at NIST for up to 1,000 s (Fig. 1c,d). All three clocks showed immunity to dominant ship motion in the band at about 0.1 Hz (Fig. 3e). A medium timescale instability was driven by the day–night temperature swing in the Conex. Nonetheless, the PICKLES–EPIC clock pair maintains $8 \times 10^{-15}$ combined instability at 100,000 s without drift correction, equivalent to temporal holdover of roughly 400 ps over 24 h. The PICKLES–EPIC data exhibit a temperature-driven instability in the $10^3$–$10^5$ s range due to insufficient air conditioner

capacity during the day. This plateau at $10^4$ s originates from EPIC on the basis of environmental chamber testing following RIMPAC, but its performance is still within two times that seen at NIST. Finally, the drift rate for PICKLES–EPIC over this period was similar to that observed at NIST (Extended Data Fig. 5). This long-term performance illustrates the robustness of iodine-based timekeeping as the clocks experienced diurnal temperature swings of several degrees, platform motion arising from ship dynamics and constant movement through Earth's magnetic field.

VIPER exhibits a short-term instability of $1.3 \times 10^{-13}/\sqrt{\tau}$ as well as a more prominent diurnal temperature instability that peaks at $4 \times 10^{-14}$ near 40,000 s (corresponding to roughly 1 day periodic instability). The VIPER physics package is an earlier design with relaxed performance goals that results in a larger temperature coefficient than the other two clocks. Nonetheless, this system can average over the diurnal temperature fluctuation and maintain an instability of $2.5 \times 10^{-14}$ after 1 day of averaging. VIPER showed a drift rate similar to PICKLES and EPIC during the underway. Importantly, the VIPER physics package

does not include magnetic shields yet still provides excellent frequency stability despite motion through Earth's magnetic field.

Summary data for PICKLES, the highest-performing clock at NIST and at sea, are shown in Fig. 1c,d. Single-clock performance at sea comprises the decorrelated instability for $\tau$ less than 200 s (Fig. 3d: blue trace) and the PICKLES–EPIC data for longer periods (Fig. 3d: black trace). The PICKLES–EPIC data are normalized by $1/\sqrt{2}$ as an upper bound for PICKLES, assuming equal contributions. Notably, the performance of PICKLES is largely unchanged at sea.

All three clocks were colocated for the at-sea testing; therefore, there is potential for correlated environmental sensitivities due to ship dynamics, motion in Earth's magnetic field and temperature and humidity variations inside the Conex. Standard reference clocks (such as a caesium beam clock or GPS-disciplined rubidium) were not available for comparison. However, simultaneous evaluation of three clocks raises the level of common mode rejection required to mask fluctuations common to the three systems, particularly given VIPER's differing spectrometer and laser system designs. Pairing the at-sea test data of three clocks with environmental testing on land provides confidence that potential correlations are below the measured instability (Supplementary Information).

Iodine has proven to be a capable platform for the development of practical optical timekeeping systems. The unique combination of SWaP, phase noise, frequency instability, low environmental sensitivity and operability on moving platforms distinguishes the approach from both commercial microwave clocks and higher-performing laboratory optical clocks. It compares favourably to active hydrogen masers in terms of long-term holdover while outperforming maser phase noise and instability at short timescales. To deliver peak performance, masers typically operate in large (approximately 1,000 l) environmental chambers that carefully regulate the temperature and humidity, limiting their use to the laboratory. Conversely, no special measures were taken to control the operating environment of the iodine clock at both NIST and throughout the RIMPAC underway. Similar to caesium beam clocks, the 3 U rackmount form factor lends itself to use outside the laboratory.

To our knowledge, these clocks are the highest-performing sea-based clocks until now. The integration, packaging and environmental robustness required to achieve such operation is a significant technological step towards widespread adoption of optical timekeeping. Since these field demonstrations, further advancement in the performance and SWaP of the rackmount clocks has been accomplished in our next-generation system, including decreasing short-term instability to $2 \times 10^{-14}/\sqrt{\tau}$, lowering the overall system SWaP to 30 l, 20 kg and 70 W and eliminating the external power supply.

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

# Methods

### System design

A block diagram for the optical clock system illustrating the physics package and electronics control system is shown in Extended Data Fig. 1a. Extended Data Fig. 1b illustrates the 2.5 l spectrometer housings that are used for the PICKLES and EPIC systems, shown schematically in Extended Data Fig. 2. The 532 nm laser system (bottom) and 1,550 nm frequency comb (top) are seen in Extended Data Fig. 1c. The combination of the laser system and frequency comb optics packages occupy a total volume of 1.5 l.

### Spectrometer

We interrogate the iodine $a_{10}$ hyperfine feature of the R(56) 32-0 transition by means of modulation transfer spectroscopy[17]. The vapour cell in which the spectroscopy is performed resides on an optical bench, and light is delivered to this bench by means of two fibre-coupled collimators. The pump and probe beams undergo several passes through the vapour cell to produce SNR capable of supporting less than $1.0 \times 10^{-13}/\sqrt{\tau}$ performance in a compact bench. Light is sampled on the bench before the cell for implementing RAM stabilization. Pump and probe powers are also actively stabilized.

The PICKLES and EPIC spectrometers use a traditional vapour cell that includes a cold finger, which is temperature stabilized at a set-point of −5 °C to maintain iodine pressure. The measured temperature-induced self-collisional frequency shift is about $-2 \times 10^{-12}$ per degree Celsius, in line with values from the literature[13]. Thus, the cold finger must be temperature stabilized at millikelvin levels to support a flicker floor of $10^{-15}$. This is accomplished with a thermo-electric cooler that surrounds the cold finger and has been able to maintain temperature fluctuations less than 1 mK for time periods exceeding 10 days, which is verified with out-of-loop thermistors colocated in the cooler structure. The VIPER spectrometer uses a starved iodine cell, which carries a fixed quantity of gaseous iodine corresponding to a fill temperature of 0 °C (roughly 4 Pa)[37,38]. This cell type eliminates the cold finger and removes the temperature dependence of the vapour density. As a result, the temperature coefficient is reduced by at least ten times, easing the cell temperature stabilization requirement. Out-of-loop temperature fluctuations were maintained to less than 10 mK throughout RIMPAC.

All optical elements routing the pump and probe beams on the optical bench are actively aligned and bonded during the build process. The result is an alignment-free subsystem. The bench containing the routing optics and vapour cell is enclosed in a thermal shield, which is also temperature stabilized. The thermal shield maintains temperature fluctuations of less than 1 mK on the optical bench for time periods exceeding 10 days, which is verified with out-of-loop thermistors located in the bench. Two of the spectrometers (PICKLES and EPIC) are packaged into 2.5 l modules, which are surrounded by a single-layer magnetic shield. VIPER uses a smaller vapour cell and has relaxed thermal control requirements, resulting in a 1 l spectrometer module meant to target more modest frequency instability metrics. VIPER also does not include a magnetic shield. Light is introduced into each spectrometer with the two fibres, and all relevant error signals exit on electrical cables.

To understand the shielding requirements, the magnetic field sensitivity of the clock transition was explored, and a second-order Zeeman shift of magnitude less than $10^{-15}$ G$^{-2}$ was observed, which is $10^8 \times$ smaller than for the 6.8 GHz rubidium microwave transition[39] and $10^4 \times$ smaller than for the rubidium two-photon transition[26]. A linear Zeeman shift less than $10^{-14}$ G$^{-1}$ is also observed for a field direction parallel to the laser beam propagation, which arises from an imperfect linear polarization of the pump beam. The singlet character of the molecular iodine clock transition provides an advantage compared to the relatively large alkali magnetic sensitivity.

### Laser system

The iodine clock transition is compatible with industry-developed 1,064 nm laser technology. The entire laser system is fiberized, permitting the use of telecom-packaged components. The laser system uses a low-noise, 1,064 nm source, and a portion of the 1,064 nm light is passed to the frequency comb. The remaining infrared light is frequency-doubled and then split to create the pump and probe beams. An acousto-optic modulator (AOM) is used to frequency offset the pump beam from the probe beam by 200 MHz to avoid a coherent background signal caused by spurious reflections[40]. In addition to the frequency offset, the AOM is used to frequency modulate the pump beam and add a controlled amplitude modulation to cancel RAM detected at the spectrometer. The frequency- and amplitude-modulated AOM drive tone is generated digitally by the field-programmable gate array control system. The modulation frequency and frequency deviation are unique to each system, but typical values are 300 kHz and 1 MHz, respectively. Laser components are spliced together to create a robust, all-fibre system from the laser front-end to the spectrometer. The assembled laser system occupies a volume of 1 l. Several different 1,064 nm lasers have been evaluated for the laser front-end with similar observed short-term instabilities for the clock.

The light shift has been measured to be approximately $10^{-12}$ mW$^{-1}$, requiring a power servo with relatively modest stability of roughly 1,000 ppm to reduce the contribution to a level of $10^{-15}$. This differs from clocks using a two-photon optical clock transition, which typically demand at least an order of magnitude tighter control over power variations[26,41].

The frequency shift with respect to RAM has been measured to be on the order of $10^{-15}$ ppm$^{-1}$, which is consistent with the clock transition linewidth of 1 MHz (including pressure and power-broadening). Stabilizing the RAM to parts-per-million levels requires state-of-the-art control but is achievable[42]. Furthermore, frequency shifts from beam-pointing errors, modulation amplitude dependence, demodulation phase and electronic baseline stability were characterized and do not presently limit clock stability.

### Frequency comb

The all-fibre frequency comb uses erbium fibre technology and is related to the design described in ref. 43. The comb oscillator is centred at 1,560 nm and operates at a repetition rate of 200 MHz. The pulse train output from this oscillator is amplified and broadened to both self-reference the comb and provide an extension to the clock subharmonic wavelength at 1,064 nm. The derived carrier-envelope offset linewidth is typically 250 kHz full-width at half-maximum with a beatnote SNR of roughly 45 dB (300 kHz resolution bandwidth). Conversely, the optical beatnote has a SNR of 40 dB (300 kHz RBW). The carrier-envelope offset beat frequency is stabilized by tuning the pump current, and the optical beatnote is stabilized with a high-bandwidth piezoelectric actuator ($f_{3dB}$ bandwidth of 200 kHz) that is incorporated into the oscillator[43]. Measurement of the timing jitter between two identical frequency combs shows that the combs can support a clock readout of $4 \times 10^{-17}$ at 1 s of averaging. The comb optics package occupies 0.5 l. Finally, conversion of the clock optical frequency to radio frequency output is accomplished with a microwave divider that converts the repetition rate to outputs at 100 MHz, 10 MHz and 1 pulse per second.

### Clock bring-up and operation

A software control system was developed that autonomously brings the system from an off state to fully locked. Following power-up, the control system sets and monitors the temperature of all critical sub-components. Once the system has reached target temperatures, a machine learning algorithm identifies the correct hyperfine manifold and centres the laser on the appropriate hyperfine transition to lock

the laser. The control system also engages the RAM and power servos for the pump and probe beams. The automated routine then locks the carrier-envelope offset frequency of the comb to a precalibrated frequency and identifies the correct comb tooth for the optical lock before stabilizing the comb to the clock laser. Once the bring-up sequence is complete, the controller continually monitors the system state for any faults and takes the appropriate action to restore the system to a fully locked state.

## NIST test setup

Extended Data Fig. 3a is a photograph of the PICKLES and EPIC systems during the NIST measurement campaign. Both systems were placed on an optical table without any further environmental shielding. Each of these systems has its own 1 U rackmount power supply (not visible) and control laptop. The 10 MHz microwave output from each clock was compared against a 5 MHz tone from the NIST maser ST05 with a Microchip 53100A phase noise analyser in a TCH configuration (Extended Data Fig. 3b and seen in the centre of the photograph). The 1,064 nm optical beatnote between the PICKLES and EPIC systems was also monitored for cross-validation. Once installed, the clocks and measurement apparatus operated autonomously. The system telemetry was remotely monitored without intervention from our California office.

A representative measurement is shown in Extended Data Fig. 3c. The raw PICKLES–EPIC instability exhibits short-term performance of $8 \times 10^{-14}/\sqrt{\tau}$. Data collection for roughly 1 day permits reliable estimation of the clock performance at 10,000 s, and noise averaging down to $2 \times 10^{-15}$ is observed. Decorrelation of the three clocks using the TCH methodology shows PICKLES and EPIC short-term instabilities of $5 \times 10^{-14}/\sqrt{\tau}$ and $6 \times 10^{-14}/\sqrt{\tau}$, respectively, outperforming the short-term performance of the ST05 maser. Excellent agreement is observed between the microwave and optical beatnotes (not shown).

The microwave phase noise between PICKLES and EPIC (Fig. 1b) was measured with the 53100A phase noise analyser, and the phase noise for a single iodine clock was inferred by subtracting 3 dB from the measured result. In particular, the phase noise at a 1 Hz offset for the 10 MHz (100 MHz) output is −124 (−105) dBc Hz$^{-1}$ before decreasing to a white phase noise floor at −170 (−155) dBc Hz$^{-1}$. More specifically, the measured phase noise $\mathcal{L}(f)$ at both frequencies may be fit to the form $\mathcal{L}(f) = \frac{1}{2} \sum_{n=-2}^{0} 10^{b_n} \times f^n$, where the coefficients $b = [b_{-2}, b_{-1}, b_0]$ correspond to white frequency, flicker phase and white phase noise, respectively[44]. The observed phase noise coefficients at 10 MHz are $b^{10\mathrm{MHz}} = [-12.2, -13.2, -16.7]$, whereas those at 100 MHz are $b^{100\mathrm{MHz}} = [-10.2, -11.9, -15.2]$. The phase noise at both frequencies is lower than commercial atomic-disciplined oscillators, highlighting the benefits of optical frequency division where the fractional noise of the iodine-stabilized laser is transferred to the radio frequency comb repetition rate.

Similarly, the optical phase noise at 1,064 nm displayed in Fig. 1b is measured between the PICKLES system and a higher-performing iodine clock. The phase noise at a 1 Hz offset is 23 dBc Hz$^{-1}$, and the measured phase noise $\mathcal{L}(f)$ follows the form described above with the coefficients $b^{1064\mathrm{nm}} = [26, 0, 0]$ down to the servo bump, which is visible at roughly 10 kHz.

## RIMPAC test setup

The experimental block diagram describing the measurements during RIMPAC 2022 is shown in Extended Data Fig. 4a. The 19-inch rack containing PICKLES, EPIC and VIPER includes a 53100A phase noise analyser to compare the 100 MHz tones derived from each clock's frequency comb against the other two in a TCH configuration. As with the NIST setup, the three pairwise optical beatnotes between the three systems were collected for cross-validation (not shown).

Frequency time traces for the three pairwise comparisons at 100 MHz are shown in Extended Data Fig. 4b. For consistency, all the collected data are shown following the VIPER external power supply restart.

As described in the main text, the VIPER physics package is an earlier design with relaxed performance goals. A diurnal temperature variation is evident in the VIPER system. Nonetheless, VIPER outperforms clocks conventionally used at sea.

## Long-term drift characterization

Extended Data Fig. 5 shows the progression of the relative PICKLES–EPIC fractional frequency, measured by means of optical beatnote at NIST, at sea for RIMPAC and at our California headquarters. The trendline represents the linear fractional frequency drift of $-2 \times 10^{-15}$ per day that was observed during the 34-day NIST measurement and illustrates several interesting features. First, the relative drift during the RIMPAC underway is indistinguishable from the NIST drift. Second, the data from these three measurements fall roughly along the trendline extrapolated from the NIST drift. This indicates that the observed long-term drift does not depend on the on–off state of the clocks, consistent with the drift origin arising from long-term changes in the vapour cell. An in-house measurement of the helium collisional shift along with calculations of the helium permeation time constant for vapour cells of our geometry indicates a fractional frequency drift of $1-2 \times 10^{-15}$ per day for 1–2 years after filling. Finally, consistency between the NIST extrapolation and the RIMPAC and California data illustrates the reproducibility and robustness of the integrated system. The time periods separating these measurements consisted of ground transport of the clocks between Boulder, CO and California and then two air-freight shipments between California and Pearl Harbor, HI. Additionally, dedicated retrace measurements of the VIPER clock at our California facility indicate frequency reproducibility less than $2 \times 10^{-14}$ following a 4 hour off state.

## Data availability

Raw and processed data generated during this study are available from the corresponding author on request.

## Code availability

Data analysis routines used to process data and generate plots are available from the corresponding author on reasonable request.

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

**Acknowledgements** A. Dowd and A. Rakholia supported electronics and code development. M. Ledbetter performed helium collision measurements, and E. Oelker carried out magnetic field evaluations. A. Vitouchkine helped design the VIPER spectrometer. J. Kohler and M. Cashen supported RIMPAC planning and operation and provided environmental data collected during the underway. L. Hollberg provided useful feedback on iodine cells and spectroscopy. J. Sherman set up the maser link and provided UTC(NIST) ensemble data, and L. Sinclair and N. Newbury supported the NIST clock measurements. T. Willis at the Office of Naval Research coordinated the US effort for the Alternative PNT Challenge at RIMPAC under The Technical Cooperation Program. Special thanks to the US Navy (NIWC Atlantic, NSWC) and the crew of the HMNZS Aotearoa for their warm hospitality and support of the at-sea demonstration. This research was developed with funding from Army Research Laboratory contract W911NF19C0047 (EPIC); Defense Advanced Research Projects Agency contracts 140D0420C0001 (HIPPOS), HR00111990037 (DILL PICKLES) and HR001121C0175 (PRICELESS); and Naval Air Systems Command contract N6833520C0116 (VIPER). The views, opinions and/ or findings expressed are those of the authors and should not be interpreted as representing the official views or policies of the Department of Defense or the US Government.

**Author contributions** J.D.R., A.C., W.D.L., G.B.P., A.S.K., D.B.S. and M.M.B. designed and built the iodine spectrometer, laser system and frequency comb. J.D.R., A.C., W.D.L., G.B.P. and A.S.K. operated the systems during field deployments. F.R. and M.M.B. developed the control system and chassis. G.E.S. and J.D.R. designed electronic subsystems for the physics packages. J.P.S. and A.C. wrote firmware for the controllers, and W.D.L. wrote the system-level automation software. J.R.A.-S. and M.M.B. supervised the project.

**Competing interests** The authors declare no competing interests.

**Additional information**
**Correspondence and requests for materials** should be addressed to Jonathan D. Roslund or Martin M. Boyd.

**A)**

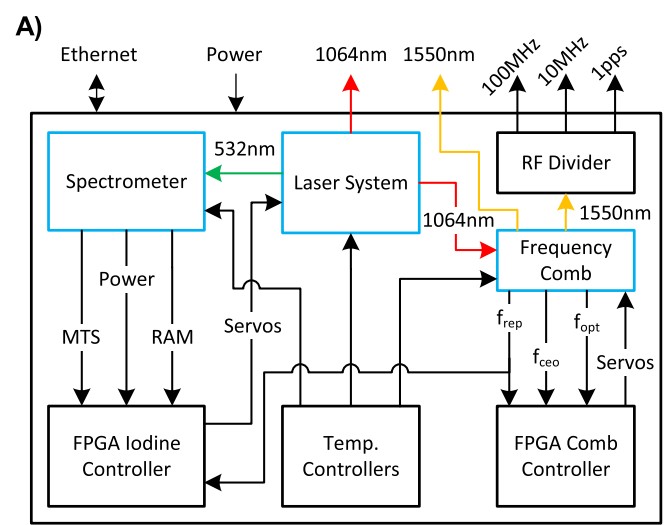

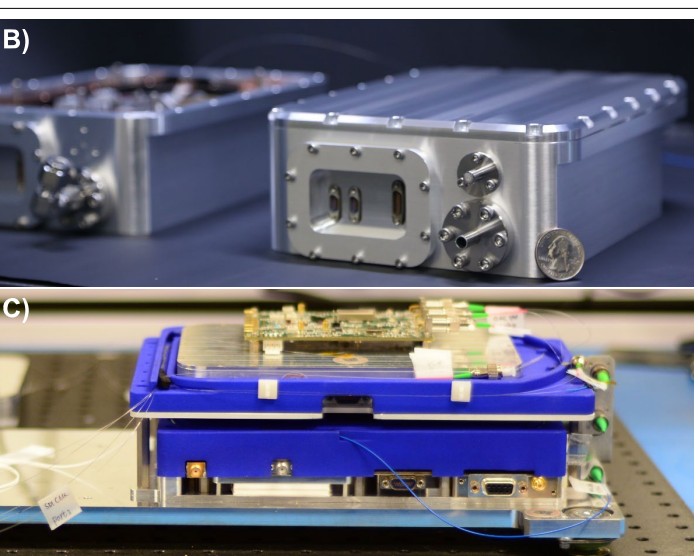

**Extended Data Fig. 1 | System block diagram and physics packages.**
A) Block diagram of the clock illustrating the physics package subsystems and the control system interfaces within the chassis. The clock chassis includes a custom (B) spectrometer, (C) laser system and fiber frequency comb optics packages (bottom and top of image, respectively).

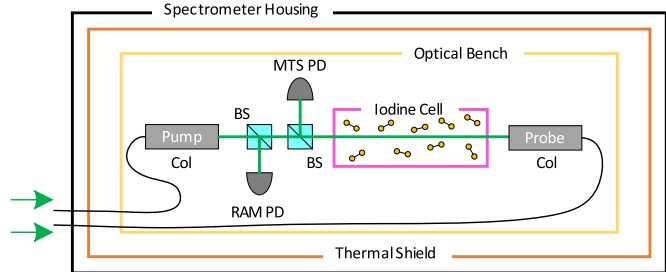

**Extended Data Fig. 2 | Iodine spectrometer schematic.** The iodine vapor cell and optics are bonded to a common optical bench. Iodine vapor density is controlled via a temperature-stabilized a cold finger. Pump and probe beams are delivered via optical fibers to free-space collimators (Col) for MTS. MTS and RAM signals are monitored on photodiodes (PD). The optical bench is enclosed in a thermal shield that is also temperature stabilized to maintain long-term sub-mk stability. For additional environmental isolation, the spectrometer housing is also temperature stabilized. A Mu-metal shield attenuates ambient magnetic fields by greater than 10×. The VIPER spectrometer operates without the cold finger and magnetic shield.

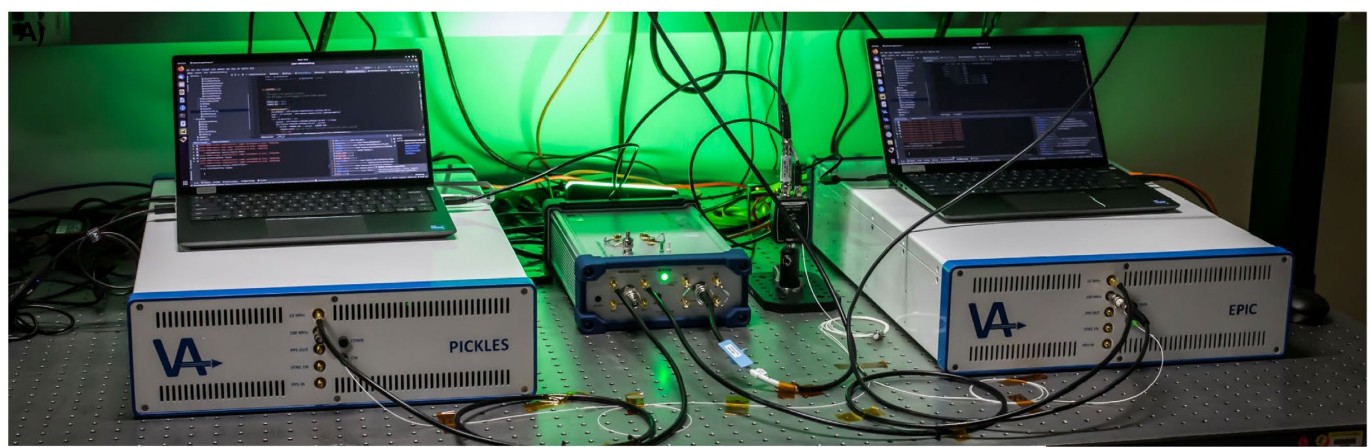

**B)**

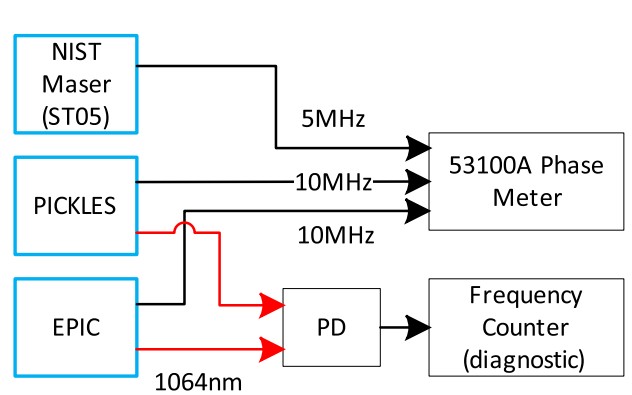

**C)**

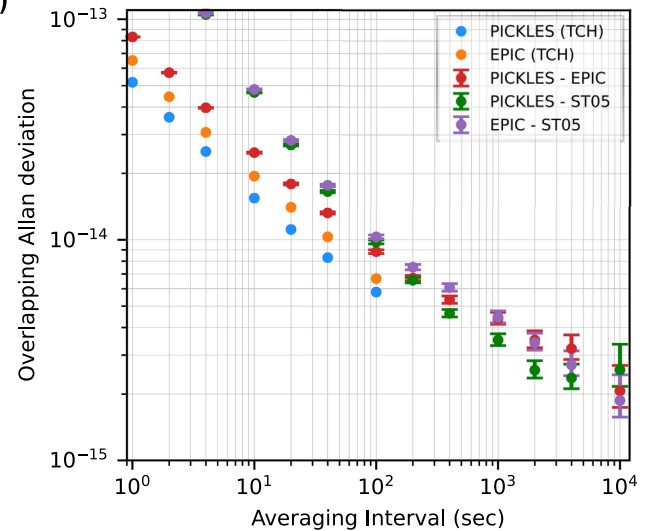

**Extended Data Fig. 3 | Clock characterization at NIST.** A) Two independent 3U, 19-inch rackmount optical clocks. Front panel outputs are 10 MHz, 100 MHz, and 1 PPS signals in addition to stabilized 1064 nm clock and 1,550 nm comb light. B) Block diagram for the measurement of two iodine clocks versus the ST05 maser at NIST Boulder. The 10 MHz outputs from each clock were compared to the 5 MHz output from ST05; diagnostics were included to monitor the 1064 nm beatnote in parallel. C) Raw instability results for the three pairwise microwave comparisons over the course of 24 h. These 1-day subsets show both clock instabilities at $\sim 2 \times 10^{-15}$ after $10^4$ seconds of averaging. Three cornered hat (TCH) extraction of the individual clock instabilities show that the two Iodine clocks operate at $5 \times 10^{-14}/\sqrt{\tau}$ and $6 \times 10^{-14}/\sqrt{\tau}$.

**A)**

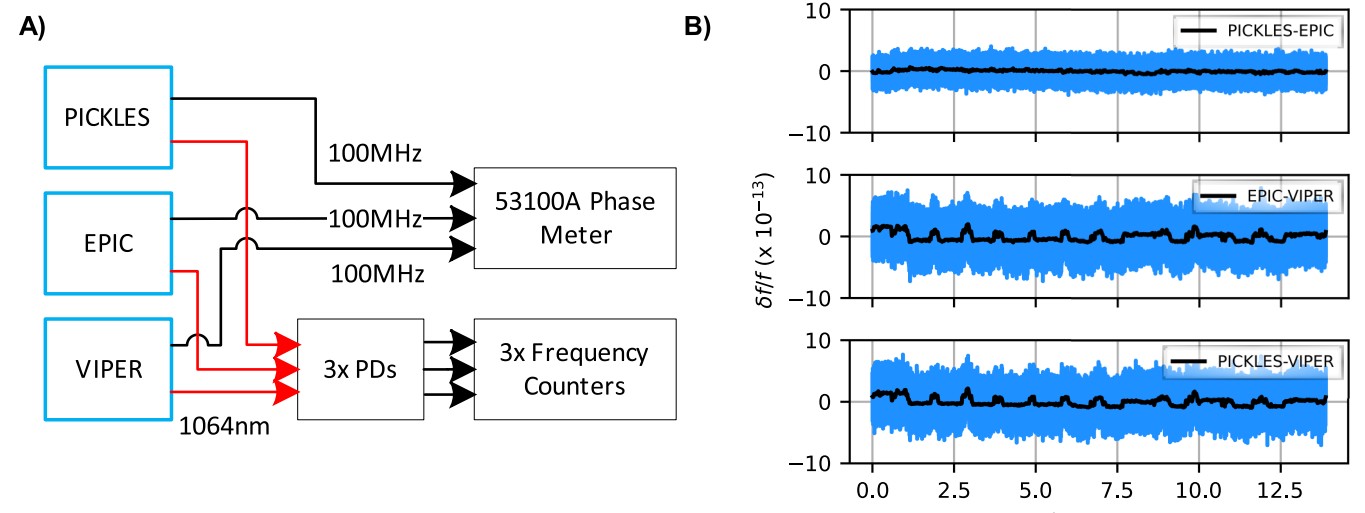

**B)**

**Extended Data Fig. 4 | Clock comparison at sea.** A) Block diagram for the measurement of three iodine clocks during RIMPAC 2022. The 100 MHz output from each clock was input to a Microsemi 53100A phase noise analyzer in a three-cornered hat configuration. The three pairwise optical beatnotes at 1064 nm were also collected in parallel. B) Time series for the three pairwise comparisons at 100 MHz over fourteen days at sea. The blue trace in each panel is the fractional frequency noise with a gate time of 1 s. The black trace is a 1,000 s moving average.

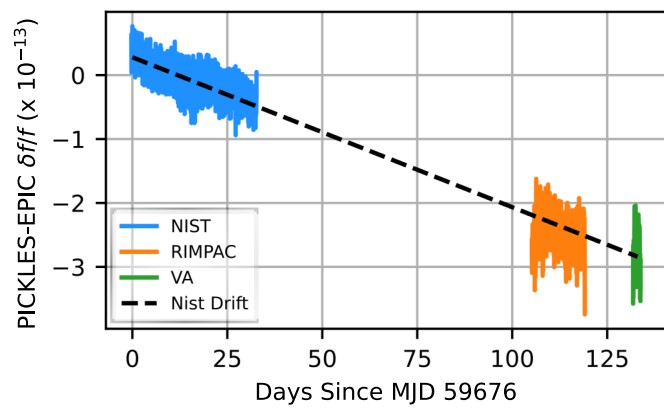

**Extended Data Fig. 5 | Long-term monitoring of PICKLES-EPIC drift (100 s moving average).** Periodic measurements at multiple sites over more than 100 days fall on the same trendline.