## [Peer Review File · Nature]

Manuscript Title: Optical Clocks at Sea

Reviewer Comments & Author Rebuttals

Reviewer Reports on the Initial Version:

Referees' comments:

Referee #1 (Remarks to the Author):

- A. This manuscript describe some exciting achievements in terms of the highest ever clock sensitivity reached at sea. The authors use iodine spectroscopy optical clocks to achieve a performance comparable to or better than the best maser systems, but in a much smaller package, using much less power and being significantly more robust.
- B. The work is original, as it for the first time realises a highly robust clock package, which works at sea with unprecedented precision. This is very exciting, as it lays the foundation for novel uses of clocks, e.g. in distributed networks with mobile components.
- C. The approach taken is solid and the presented data is gained in well described and suitable procedures. The NIST data clearly demonstrates the superior performance of the clocks over the maser systems there. The data at sea is taken in a suitable way to reference the clocks against each other. There is only one question, which I would like to see discussed: Is there a possibility that the performance measurement at sea is compromised by common reactions of the clocks to the environment? As all the clocks are housed in the same rack, experiencing the same temperature changes, rotations and accelerations, one wonders whether this might influence the deductions of clock instability in a way making the clocks look better than they are, if they all had the same types of frequency shifts,...
- D. The use of statistics and treatment of uncertainties seems fine to me.
- E. The conclusions are well presented and seem valid. I just would like to see the discussion of possible common deviations at sea.
- F. Suggested Improvements: See above. In addition, it would be nice to see some schematics of the iodine spectroscopy setup, but I appreciate that this might need to be kept commercially in confidence.
- G: The references provide adequate credit to previous work.
- H. The manuscript is extremely well written and gives a clear context. The abstract, introduction and conclusion are fully appropriate and meaningful to the non-expert.

Referee #2 (Remarks to the Author):

- A. Summary: This article describes the design of a fieldable, integrated, iodine optical clock that provides high performance RF and optical frequency outputs. It presents the performance of several iodine optical clock systems continuously operated at NIST for ~34 days, and at sea onboard a navy ship for ~20

days. The results are impressive, showing $< 10E-14$ frequency stability and the ability to maintain sub-ns time in the challenging shipboard environment.

This work represents a significant contribution to the advancement of navigation and global time distribution, especially in support of operations without dependency on GNSS and enabling moving platforms to access optical quality performance. The article provides a clear motivation for the development, describes the key elements of the system and the testing methods used at NIST and onboard the ship. A key highlight is that the operation is entirely automatic with minimal environmental controls.

B. The methods and data look valid and appropriate.

F.

I have a few questions:

- Differences between clocks (PICKLES and EPIC). On line 51 it describes these two clocks as identical, but later on 96-101 it describes an issue with an etalon on PICKLES where this part had been removed from EPIC. I'm not clear on how big a difference this represents, or what was used to replace this component. I don't understand the related comment (line 101): "measures have been taken to reduce it in future versions of the clock."
- Line 145 – It states that EPIC presents a temperature-driven instability in the $1E4 - 1E5$ s range at sea, but I don't see this in the figures unless it is the plateau shown for the unnamed clock in Figure 1C; but even that seems to start at a shorter interval than $1E4$. More information about this comment would be useful (147).
- Line 148 "Finally, the drift rate for PICKLES-EPIC over this period was similar to that observed at NIST." I don't see any plots of the drift rate for PICKLES-EPIC at sea. It would be useful to include for comparison with the inset of Figure 2 from the NIST tests.
- Line 159-160 This comment at the end "Summary data for the highest performing clock at NIST and at sea is shown in Figure 1C and D" could use more explanation. Is the same "best clock" shown in all cases or is a different clock shown in the two graphs? It would be useful to include the name so that it can be connected with the other figures in the paper.

G. References - Please check on Reference 35 - I am not sure if it is a publicly available document.

H. The entire text, including the abstract, is clear, well written, and appropriate.

Referee #3 (Remarks to the Author):

This is a well written paper that describes for the first time the operation of optical clocks (in this case a specific type based on the Doppler-free spectroscopy of molecular iodine in a vapor-cell) on a ship at sea. The data presented for the naval measurement campaign, as well as for the preceding characterization measurements, is of very high quality and fulfils the highest standards with respect to measurement methodology and data analysis. However, while the work is certainly novel, interesting and deserves publication in some form, I don't consider it to be of sufficient interest to a wide enough audience to warrant publication in nature itself.

Concerning the publication in another journal (NATURE communications comes to mind), I actually find it quite difficult to make a clear recommendation for to the following reason:

Quite obviously in this manuscript some important technical information is not provided as I would expect (see below for details), which I suspect to be for confidentiality reasons. If one adheres to the principle that a knowledgeable reader with sufficient resources should in principle be able to replicate the results presented in a publication, then those omissions would likely be contrary to that. Thus, if the manuscript had been submitted by a "normal" research group located at a university of publicly funded research lab, then I would recommend rejecting the manuscript unless it is revised to provide many more details or at least cite an existing reference where those details are given. But as this comes from a commercial company, that leads to a somewhat different perspective.

In the end, the journal will have to weigh the various arguments in order to come to a decision in this special case. Here I will just provide my detailed comments to provide input for the discussion and motivate the authors to provide as much additional information as possible.

First on the above-mentioned technical details:

Line 49-50 "The integrated control system is a mix of custom and commercial circuit boards, with custom firmware."

 analog or digital, FPGA? In Figure 4A it can be found that the control electronics is based on FPGA. Some more detail on the implementation of the lock electronics should be given. Active power stabilization, RAM control could be mentioned here.

Line 100 "This etalon was removed during the building of the EPIC spectrometer"

 At this point it is not clear if there is any RAM control. Would be interesting to know where RAM is detected. Before iodine cell? Where is the pickup? Same questions for the optical power stabilization.

Line 131-132 "Operation in dynamic environments highlights the robust, high-bandwidth clock readout enabled by a vapor cell."

 What is the control bandwidth of the frequency stabilization of the 1064 nm laser?

Line 163 "SWaP"

 What are the SWaP budgets of the Systems (EPIC, VIPER, PICKELS)?

Line 204 "Spectrometer"

 Does the spectrometer apply balanced detection? Does/would it help to increase SNR?

Line 208 – 209 "racetrack configuration" / "adequate"

 What is meant by racetrack configuration here? What is the length of the cell and the number of passes / total path length? What is considered adequate? A schematic of the optical layout of the spectrometer would be appreciated.

Line 212 "a setpoint below 0 °C"

 How far below 0°C / at what temperature?

Line 218 "vapor pressure at the chosen fill temperature"

 To what pressure is the cell filled?

Line 239 "laser technology"

 What kind of laser technology is meant here? Fiber, diode, solid state? This might also be interesting to estimate the impact of frequency noise (inter-modulation noise) and intensity noise on the SNR.

Line 243 "The pump beam is frequency offset from the probe beam"

 What is the frequency offset (with respect to Doppler broadening...)? What frequency source is used to generate the offset frequency with sufficient stability?

Line 245 "AOM is used to frequency modulate (FM)"

 Could you please provide information on modulation frequency, modulation index, optical powers ... ?

Line 247-248 "single spool design"

 What exactly is a single spool design and how does it contribute to robustness compared to, e.g., a multi spool design?

Line 256-258 “Furthermore, frequency shifts from beam pointing errors, modulation amplitude dependence, de-modulation phase, and electronic baseline stability were quantified, and appropriate measures were implemented to minimize their contribution.”

 Very vague. Could you please give you give the coefficient and contribution to the instability of the spectrometer for each effect?

Line 259 “Frequency Comb”

 How is the comb locked to the CW laser. Is it an all optical locking scheme or is CEO frequency locked to some other RF reference?

Line 283 / Figure caption (A)

 Power stabilization seems to be missing in the block diagram, i.e., arrow(s) between spectrometer and FPGA Iodine controller

Line 283 / Figure caption (B)

 Are the spectrometers hermetically sealed? Filled with a specific gas mixture? Evacuated?

Then on other things:

Title “Optical Clocks at Sea”

 seems to be overly broad, “Optical Iodine Clocks at Sea” might be more appropriate.

Line 18 “Operating high-performance clocks at sea has been historically challenging”

 “historically challenging” here might reference a bit too far back in time ( Harrison ?). For mechanical and even quartz clocks a ship might indeed be an especially challenging environment due to its motion, but does this still hold true for, e.g., atomic cesium or rubidium clocks? It would be better to instead present (although not in the abstract) a comparative overview of the challenges for mobile clocks in various environments (surface ship, submarine, spacecraft, fixed wing aircraft, helicopter, automotive, ...)

Line 32 “first demonstrations of optical clocks [19].”

 should also mention Nevsky, 10.1016/S0030-4018(01)01190-7

Line 44 “shown in Figure 1A”

 Figure 1A does not really show clock prototypes, but rather a clock chassis.

Sentence could read instead: Initial clock prototypes were integrated into 35 L, 3U 19-inch rackmount chassis, shown in Figure 1A.

Line 52-53 “A third clock (VIPER) was built using a smaller iodine spectrometer and laser system to reduce physics package SWaP at relaxed performance goals.”

 How much was S W and P reduced? By what measures and what are the size, the weight, and the power consumption of the clocks? Later in the text, only the SWaP of a new development is mentioned. Further, later in the text it is written that VIPER was built before PICKLES and EPIC... i.e., the SWaP was increased and the performance improved...

Line 55 “Results”

 From the measurements performed with the clocks, can you provide information on the frequency reproducibility, retrace of the clocks?

Line 58 “operated on an optical table without any special environmental measures”

 No temperature stabilization at NIST Labs? It is later stated that: The laboratory housing PICKLES and EPIC was stable to 100's of mK throughout the measurement campaign. Standard temperature stabilization could be mentioned here.

Line 80-82 “the drift rates ...”

 The drift rate looks surprisingly linear for such a long duration. Are there any ideas about the cause of the linear drift? Comparison of the linear drift to the accuracy of another iodine clock is not plausible.

Line 86-88 “Without drift removal, the long-term clock performance is competitive with active hydrogen masers; drift removal puts the clock stability on-par with the highest performing masers.”

 This is statement should be reconsidered, the second part is definitely not true (see e.g., <https://www.vremya-ch.com/index.php/en/products-en/activehm-en/vch-2021-en/index.html>). In general, Masers still seem to be better in the especially application relevant range between 1E4 and 1E6 seconds.

Line 96-97 “7 hour timescales”

 To find the peak deviation in the Allan deviation plot, where the averaging time is given in seconds, stating “20 ks (~7 h)” might help.

Line 97-98 “shift of the hyperfine transition linewidth”

 A shift of the 'linewidth' does not make physical sense. Likely here RAM introduces a shift in the lockpoint, i.e. an offset from the center of the transition, rather than a shift to the transition ... Also, the term 'hyperfine transition' could be reconsidered.

Caption Figure 3 (A) “Clock stack-up for RIMPAC 2022. The server rack contained three independent optical clocks”

 Can you elaborate on common mode rejection of environmental effects with the three clocks assembled close together in the same container? Did you consider comparison to GPS?

Line 99 “uncontrolled”

 What is a controlled etalon? Maybe “spurious” or “parasitic”?

Line 106-107 “In addition, the iodine-stabilized laser provides ...”

 Compared to other papers cavities only a factor of 1000 ?

<https://iopscience.iop.org/article/10.7567/APEX.7.022705/meta>

ULE Cavity with drift rate of 2.16 ± 0.09 kHz/d at 657 nm

The cited paper reports on the drift rate between two lasers locked to two ULE cavities which might not be a good measurement of the drift rate of an individual cavity. Maybe also move this comparison to the section on the drift results where the drift is already compared to RAFS...?

Line 125 “3°C”

 Please specify if this is peak-to-peak or amplitude.

Line 173 “To our knowledge, these clocks are the highest performing sea-based clocks to date.”

 Very carefully worded (“To our knowledge”), but with some probability there might actually be some higher performing sea-based clocks. Should check Russian and Chinese language publications (Masers ...), also follow up on <https://www.royalnavy.mod.uk/news-and-latest-activity/news/2022/march/11/220311-hms-pwls-quantum-technology>

Line 215 “oven”

 Is "oven" a good word here? Maybe a (thermo-electric) cooler?

Figure 6

 There seems to be a distinct pattern in the frequency of the VIPER clock. Is there an explanation for this behavior? Is it correlated to maneuvers of the ship?

References [37] and [38]

 could only find abstracts which do not provided details to the statements made!

Referee #4 (Remarks to the Author):

In this manuscript, the authors present a compact and robust implementation of an optical iodine clock. Iodine spectrometer, frequency comb and associated control electronics are combined in one single package, representing a near-product implementation of a frequency standard. They highlight their setups and show the measured performance, once in the lab at NIST and once during operation on a ship. The shown frequency instabilities of the iodine clock are comparable to those of an active hydrogen maser.

In my opinion, the presented implementation is very impressive regarding the obtained frequency stability in combination with compactness (35 l of volume) and robustness. Operation on a ship demonstrated the functionality under harsh environmental conditions (with respect to temperature variations and accelerations). The resulting frequency instabilities are comparable to the current state of the art in iodine references where – to my knowledge – long-term stabilities of up to 1.000.000 s have not yet been reported.

The manuscript is very well written and publication of the results is very welcome. It is recognized, that such a compact and easy-to-handle frequency standard will improve current applications of frequency standards and also open up new applications as stated in the manuscript. However, originality and relevance are not seen high enough for publication in Nature. Within Nature Portfolio Nature Communications, Light: Science and Applications, Communications Physics or Scientific Reports are considered more appropriate journals. The techniques used (iodine modulation transfer spectroscopy and Er fiber frequency comb) are well known and no novel techniques or methods are presented for the compact and robust implementation.

Going to more detail, I would like to add the following points:

1. The shown Allan deviation plots include curves with de-drifted data. What is the justification for this linear detrend? Without linear detrend of the data, the exclusiveness of the long-term stability is very much reduced.
2. Application of the iodine clock on GNSS satellites is stated but suitability and relevant delta steps for space compatibility of the implementation are not given.
3. The manuscript lacks technical details of their implementation that are of interest to the

corresponding community. These include, for example, the length of the gas cell, the interaction pathlength within the cell, the beam diameter and the intensities of pump and probe beams.

4. Since the clocks were operated in the same container (and in the same rack), they were subject to the same temperature variations. Were common-mode effects taken into account?

5. Figure 1C shows the Allan deviation of the iodine clock at NIST and at sea. What is the underlying measurement (beat measurement between PICKLES and EPIC, three-cornered hat analysis)?

6. It is stated, that the VIPER clock has smaller SWaP budgets and lower performance goals but no details on the different implementation is given. This would also be of interest with respect to the Allan deviations shown in Figure 3D.

7. While the temperature stability for the PICKLES and EPIC iodine cells' cold fingers are given, this information lacks for the VIPER starved cell. (line 220)

8. What is the temperature stability of the optical bench provided by the thermal shielding and at what absolute temperature is it operated? (line 225)

9. More details on the used 1064nm laser source would be interesting, including type of laser, output power and linewidth. (line 241)

Author Rebuttals to Initial Comments:

Referee #1 (Remarks to the Author):

Authors: We are pleased that the reviewer recognized the impact of a compact optical clock for next-generation timekeeping. Their detailed feedback has helped us to strengthen the manuscript. Care has been taken to address their feedback while staying within the word count guidelines. Where fixes were suggested, we have addressed the referees' comments. Changes are identified in-line with the referee comments below and redlined in the revised .doc file.

- A. This manuscript describe some exciting achievements in terms of the highest ever clock sensitivity reached at sea. The authors use iodine spectroscopy optical clocks to achieve a performance comparable to or better than the best maser systems, but in a much smaller package, using much less power and being significantly more robust.
- B. The work is original, as it for the first time realises a highly robust clock package, which works at sea with unprecedented precision. This is very exciting, as it lays the foundation for novel uses of clocks, e.g. in distributed networks with mobile components.
- C. The approach taken is solid and the presented data is gained in well described and suitable procedures. The NIST data clearly demonstrates the superior performance of the clocks over the maser systems there. The data at sea is taken in a suitable way to reference the clocks against each other. There is only one question, which I would like to see discussed: Is there a possibility that the performance measurement at sea is compromised by common reactions of the clocks to the environment? As all the clocks are housed in the same rack, experiencing the same temperature changes, rotations and accelerations, one wonders whether this might influence the deductions of clock instability in a way making the clocks look better than they are, if they all had the same types of frequency shifts,...

Author Response 1: Added analysis/text placing an upper bound on potential environmental correlations due to acceleration, temperature, and magnetic fields. This text is split between the main body and the Supplemental Material:

All three clocks were co-located for the at-sea testing, therefore, there is potential for correlated environmental sensitivities due to ship dynamics, motion in earth's magnetic field, and temperature and humidity variations inside the Conex. Standard reference clocks (e.g., cesium beam clock or GPS-disciplined rubidium) were not available for comparison. However, simultaneous evaluation of three clocks raises the level of common mode rejection required to mask fluctuations common to the three systems, particularly given VIPER's differing spectrometer and laser system designs. Pairing the at-sea test data of three clocks with environmental testing on land provides confidence potential correlations are below the measured instability (See Supplemental Material).

AND

The acceleration sensitivity for VIPER was measured on land at $< 1 \times 10^{-14}/g$. For timescales relevant to ship dynamics (timescales $< 10^3$ s), scaling the measured acceleration in Figure 3E by this coefficient implies that acceleration-induced shifts are $\sim 100\times$ below the clock noise. Timescales

from $10^3 - 10^5$ are more challenging to analyze and contain regions where temperature variation is strongest. Using independent environmental chamber testing on land, temperature coefficients were measured for PICKLES ($< 5 \times 10^{-15}/^\circ\text{C}$), EPIC ($2 \times 10^{-14}/^\circ\text{C}$), and VIPER ($8 \times 10^{-14}/^\circ\text{C}$). Given this, common mode temperature fluctuations between clocks are relatively small, and the correlation between the Conex temperature fluctuations and clock instabilities is consistent with the measured temperature coefficients. Finally, measured drift rates consistent with the NIST measurements suggest that the instabilities for $> 10^5$ s timescales accurately reflect the intrinsic behavior of the clocks. Notably, at the level that correlations could exist, cesium beam clocks are not precise enough to identify them on these timescales (e.g., Figure 1, Green trace).

With respect to magnetic fields, VIPER's sensitivity has been measured at $< 10^{-14}/\text{Gauss}$ despite having no shield. EPIC and PICKLES include shielding which reduces the impact of Earth's field by $> 10\times$. Given the scale of the shifts and differences in the clock hardware, correlated magnetic sensitivity during the underway is very unlikely at these performance levels. Finally, no humidity-correlated frequency instabilities were observed among the three clocks despite $\sim 15\%$ changes over the ~ 10 -minute AC period and 4-5% over day-night cycles throughout the underway.

- D. The use of statistics and treatment of uncertainties seems fine to me.
- E. The conclusions are well presented and seem valid. I just would like to see the discussion of possible common deviations at sea.

See Author Response 1.

- F. Suggested Improvements: See above. In addition, it would be nice to see some schematics of the iodine spectroscopy setup, but I appreciate that this might need to be kept commercially in confidence.

Author Response 2: Added a simplified spectrometer schematic (Figure 5) to Supplemental Material.

- G. The references provide adequate credit to previous work.
- H. The manuscript is extremely well written and provides a clear context. The abstract, introduction and conclusion are fully appropriate and meaningful to the non-expert.

Referee #2 (Remarks to the Author):

Authors: We are pleased that the reviewer recognized the impact of a compact optical clock for next-generation timekeeping. Their detailed feedback has helped us to strengthen the manuscript. Care has been taken to address their feedback while staying within the word count guidelines. Where fixes were suggested, we have addressed the referees' comments. Changes are identified in-line with the referee comments below and redlined in the revised .doc file.

- A. This article describes the design of a fieldable, integrated, iodine optical clock that provides high performance RF and optical frequency outputs. It presents the performance of several iodine optical clock systems continuously operated at NIST for ~ 34 days, and at sea onboard a navy ship for ~ 20 days. The results are impressive, showing $< 10\text{E}-14$ frequency stability and the ability to maintain sub-ns time in the challenging shipboard environment.

This work represents a significant contribution to the advancement of navigation and global time distribution, especially in support of operations without dependency on GNSS and enabling moving platforms to access optical quality performance. The article provides a clear motivation for the development, describes the key elements of the system and the testing methods used at NIST and onboard the ship. A key highlight is that the operation is entirely automatic with minimal environmental controls.

B. The methods and data look valid and appropriate.

F. I have a few questions:

- Differences between clocks (PICKLES and EPIC). On line 51 it describes these two clocks as identical, but later on 96-101 it describes an issue with an etalon on PICKLES where this part had been removed from EPIC. I'm not clear on how big a difference this represents, or what was used to replace this component. I don't understand the related comment (line 101): "measures have been taken to reduce it in future versions of the clock."

Author Response 1: Modified two passages to resolve unclear text:

Two clocks with identical hardware (PICKLES and EPIC) were developed with physics packages targeting short-term instability below $10^{-13}/\sqrt{\tau}$, comparable to commercial masers.

AND

We suspect that the origin of this plateau in PICKLES is RAM coupling through a spurious etalon in the spectrometer. By modifying the build procedure, this etalon was mitigated during the build of the EPIC spectrometer.

- Line 145 – It states that EPIC presents a temperature-driven instability in the 1E4 – 1E5 s range at sea, but I don't see this in the figures unless it is the plateau shown for the unnamed clock in Figure 1C; but even that seems to start at a shorter interval than 1E4. More information about this comment would be useful (147).

Author Response 2: The reviewer correctly notes that the temperature instability is the broad plateau, seen in Figure 3D (black trace) for the at-sea data. The same low-Q peak is seen in the red trace for VIPER, except that the temperature coefficient is 4-5x larger. Clarified the text by updating the time range and including a parenthetical note:

The PICKLES-EPIC data exhibits a temperature-driven instability in the $10^3 - 10^5$ s range due to insufficient air conditioner (AC) capacity during the day. This plateau at $\sim 10^4$ s originates from EPIC based on environmental chamber testing following RIMPAC, but its performance is still within $2\times$ of that seen at NIST.

- Line 148 "Finally, the drift rate for PICKLES-EPIC over this period was similar to that observed at NIST." I don't see any plots of the drift rate for PICKLES-EPIC at sea. It would be useful to include for comparison with the inset of Figure 2 from the NIST tests.

Author Response 3: Added a plot of the measured drift over time for PICKLES-EPIC at NIST, RIMPAC, and our California facility (Figure 8) and corresponding text to Supplemental Material:

Figure 8 shows the progression of the relative PICKLES-EPIC fractional frequency, measured via optical beatnote at NIST, at-sea for RIMPAC, and at our California headquarters. The trendline represents the linear fractional frequency drift of $\sim -2 \times 10^{-15}$ /day that was observed during the 34-day NIST measurement and illustrates several interesting features. First, the relative drift during the RIMPAC underway is indistinguishable from the NIST drift. Second, the data from these

three measurements approximately fall along the trendline extrapolated from the NIST drift. This suggests that the observed long-term drift does not depend upon the on-off state of the clocks, consistent with the drift origin arising from long term changes in the vapor cell. An in-house measurement of the helium collisional shift along with calculations of the helium permeation time constant for vapor cells of our geometry suggest a fractional frequency drift of $\sim 1-2 \times 10^{-15}$ /day for 1-2 years after filling. Finally, consistency between the NIST extrapolation and the RIMPAC and California data illustrate the reproducibility and robustness of the integrated system. The time periods separating these measurements consisted of ground transport of the clocks between Boulder, CO and California and then two air-freight shipments between California and Pearl Harbor, HI. Additionally, dedicated retrace measurements of the VIPER clock at our California facility indicate frequency reproducibility $< 2 \times 10^{-14}$ following a 4-hour off state.

• Line 159-160 This comment at the end “Summary data for the highest performing clock at NIST and at sea is shown in Figure 1C and D” could use more explanation. Is the same “best clock” shown in all cases or is a different clock shown in the two graphs? It would be useful to include the name so that it can be connected with the other figures in the paper.

Author Response 4: Expanded the sentence into a new paragraph for clarity:

Summary data for PICKLES, the highest performing clock at NIST and at sea is shown in Figure 1C and D. Single clock performance at sea is comprised of the decorrelated instability for $\tau < 200$ s (Figure 3D – blue trace) and the PICKLES-EPIC data for longer periods (Figure 3D – black trace). The PICKLES-EPIC data is normalized by $1/\sqrt{2}$ as an upper bound for PICKLES assuming equal contributions. Notably, the performance of PICKLES is largely unchanged at sea.

G. References - Please check on Reference 35 - I am not sure if it is a publicly available document.

Author Response 5: The referee correctly points out that there is no Conference Proceeding for this reference, only an online abstract (<https://www.ion.org/jnc/abstracts.cfm?paperID=12097>). However, this is the only reference we are aware of that captures this team’s contribution at RIMPAC; previous laboratory contributions are captured in [26] and [43]. We feel it is essential to credit their work at RIMPAC but defer to the Editor on how to handle the citation appropriately.

H. The entire text, including the abstract, is clear, well written, and appropriate.

Referee #3 (Remarks to the Author):

Authors: We are pleased that the reviewer recognized the first-of-kind effort in bringing 3x optical clocks on board a moving platform in such a compact form factor. Their detailed feedback has helped us to strengthen the manuscript. Care has been taken to address their feedback while staying within the word count guidelines. Where fixes were suggested, we have addressed the referees’ comments. Changes are identified in-line with the referee comments below and redlined in the revised .doc file.

We maintain that these optical clocks mark a significant advancement for quantum sensors owing to their combination of high performance, compact footprint, and environmental robustness. Their real-world operability can have broad societal and scientific impact for navigation, telecommunications, and astronomy. We reiterate that the impact of our work is technological

rather than fundamental. Similar work has been featured in Nature [1], where device performance is not laboratory state-of-the-art, but actually fielding the instrument can have transformative utility, including for fundamental science.

[1] Nature 602, 590–594 (2022)

This is a well written paper that describes for the first time the operation of optical clocks (in this case a specific type based on the Doppler-free spectroscopy of molecular iodine in a vapor-cell) on a ship at sea. The data presented for the naval measurement campaign, as well as for the preceding characterization measurements, is of very high quality and fulfils the highest standards with respect to measurement methodology and data analysis. However, while the work is certainly novel, interesting and deserves publication in some form, I don't consider it to be of sufficient interest to a wide enough audience to warrant publication in nature itself.

Concerning the publication in another journal (NATURE communications comes to mind), I actually find it quite difficult to make a clear recommendation for to the following reason:

Quite obviously in this manuscript some important technical information is not provided as I would expect (see below for details), which I suspect to be for confidentiality reasons. If one adheres to the principle that a knowledgeable reader with sufficient resources should in principle be able to replicate the results presented in a publication, then those omissions would likely be contrary to that. Thus, if the manuscript had been submitted by a "normal" research group located at a university of publicly funded research lab, then I would recommend rejecting the manuscript unless it is revised to provide many more details or at least cite an existing reference where those details are given. But as this comes from a commercial company, that leads to a somewhat different perspective.

In the end, the journal will have to weigh the various arguments in order to come to a decision in this special case. Here I will just provide my detailed comments to provide input for the discussion and motivate the authors to provide as much additional information as possible.

First on the above-mentioned technical details:

Line 49-50 “The integrated control system is a mix of custom and commercial circuit boards, with custom firmware.”

 analog or digital, FPGA? In Figure 4A it can be found that the control electronics is based on FPGA. Some more detail on the implementation of the lock electronics should be given. Active power stabilization, RAM control could be mentioned here.

Author Response 1: Added the following clarifying details about the control system and introduced the various controls implemented, including RAM:

FPGA-based controllers perform digital locks for the laser and frequency comb, servo residual amplitude modulation (RAM), and stabilize the pump and probe powers.

Line 100 “This etalon was removed during the building of the EPIC spectrometer”

 At this point it is not clear if there is any RAM control. Would be interesting to know where RAM is detected. Before iodine cell? Where is the pickup? Same questions for the optical power stabilization.

Author Response 2: Captured in AR 7 and added specificity to the Methods section:

Light is sampled on the bench before the cell for implementing RAM stabilization. Pump and probe powers are also actively stabilized.

Line 131-132 "Operation in dynamic environments highlights the robust, high-bandwidth clock readout enabled by a vapor cell."

 What is the control bandwidth of the frequency stabilization of the 1064 nm laser?

Author Response 3: Added the parenthetical information:

Operation in dynamic environments highlights the robust, high-bandwidth clock readout (> 10 kHz control bandwidth) enabled by a vapor cell.

Line 163 "SWaP"

 What are the SWaP budgets of the Systems (EPIC, VIPER, PICKELS)?

Author Response 4: Size is already captured in the manuscript. Added weight and power:

Each system consumes ~85 Watts (excluding the external power supply) and weighs ~26 kilograms.

Line 204 "Spectrometer"

 Does the spectrometer apply balanced detection? Does/would would it help to increase SNR?

Author Response 5: The exact detection scheme for the MTS signal is currently under review for patent and so not included here

Line 208 – 209 "racetrack configuration" / "adequate"

 What is meant by racetrack configuration here? What is the length of the cell and the number of passes / total path length? What is considered adequate? A schematic of the optical layout of the spectrometer would be appreciated.

Author Response 6: Added a simplified spectrometer schematic (Figure 5) to Supplemental Material and the clarifying language:

The pump and probe beams undergo multiple passes through the vapor cell to produce SNR capable of supporting $< 1.0 \times 10^{-13} / \sqrt{\tau}$ performance in a compact bench.

Line 212 "a setpoint below 0 °C"

 How far below 0°C / at what temperature?

Author Response 7: Added clarifying language:

The PICKLES and EPIC spectrometers utilize a traditional vapor cell that includes a cold finger, which is temperature stabilized at a setpoint of ~5 °C to maintain iodine pressure.

Line 218 "vapor pressure at the chosen fill temperature"

 To what pressure is the cell filled?

Author Response 8: Added clarifying language:

The VIPER spectrometer employs a starved iodine cell, which carries a fixed quantity of gaseous iodine corresponding to a fill temperature of ~ 0 °C (~ 4 Pascals) [39] [40].

Line 239 “laser technology”

 What kind of laser technology is meant here? Fiber, diode, solid state? This might also be interesting to estimate the impact of frequency noise (inter-modulation noise) and intensity noise on the SNR.

Author Response 9: We prefer not to identify the specific laser used, but added clarifying language to Methods:

Several different 1064 nm lasers have been evaluated for the laser front-end with similar observed short-term instabilities for the clock.

Line 243 “The pump beam is frequency offset from the probe beam”

 What is the frequency offset (with respect to Doppler broadening...)? What frequency source is used to generate the offset frequency with sufficient stability?

Author Response 10: Added specificity:

An acousto-optic modulator (AOM) is used to frequency offset the pump beam from the probe beam by ~ 200 MHz to avoid a coherent background signal due to spurious reflections [42].

AND

The frequency- and amplitude-modulated AOM drive tone is generated digitally by the FPGA control system.

Line 245 “AOM is used to frequency modulate (FM)”

 Could you please provide information on modulation frequency, modulation index, optical powers ... ?

Author Response 11: Added specificity:

The modulation frequency and frequency deviation are unique to each system, but typical values are ~ 300 kHz and ~ 1 MHz, respectively.

Line 247-248 “single spool design”

 What exactly is a single spool design and how does it contribute to robustness compared to, e.g., a multi spool design?

Author Response 12: Added clarification:

Laser components are spliced together to create a robust, all-fiber system from the laser front-end to the spectrometer. The assembled laser system occupies a volume of 1 L.

Line 256-258 “Furthermore, frequency shifts from beam pointing errors, modulation amplitude dependence, de-modulation phase, and electronic baseline stability were quantified, and appropriate measures were implemented to minimize their contribution.”

 Very vague. Could you please give you give the coefficient and contribution to the instability of the spectrometer for each effect?

Author Response 13: This manuscript reports on high-level field testing and systematics. Lower-level systematics referred to in this sentence were determined to contribute at levels below the results presented. To emphasize this point, the text has been amended to read:

Furthermore, frequency shifts from beam pointing errors, modulation amplitude dependence, demodulation phase, and electronic baseline stability were characterized and do not presently limit clock stability.

Line 259 “Frequency Comb”

 How is the comb locked to the CW laser. Is it an all optical locking scheme or is CEO frequency locked to some other RF reference?

Author Response 14: Cross-referenced [45-Sinclair] for this standard comb locking scheme:

The CEO beat frequency is stabilized by tuning the pump current, and the optical beatnote is stabilized with a high-bandwidth PZT actuator (f_{3dB} bandwidth $\sim 200\text{kHz}$) that is incorporated into the oscillator [45].

Line 283 / Figure caption (A)

 Power stabilization seems to be missing in the block diagram, i.e., arrow(s) between spectrometer and FPGA Iodine controller

Author Response 15: To aid the reader we've added an arrow indicating power stabilization to block diagram.

Line 283 / Figure caption (B)

 Are the spectrometers hermetically sealed? Filled with a specific gas mixture? Evacuated?

Author Response 16: As the referee pointed out, the manuscript does not cover all technical information necessary to replicate the result. To address this, we have added significant technical information and included several references to related clock work employing similar techniques.

This has been redacted.

We trust that the reviewer will see that we took their questions seriously and have been as transparent as possible with our responses.

Then on other things:

Title “Optical Clocks at Sea”

 seems to be overly broad, “Optical Iodine Clocks at Sea” might be more appropriate.

Author Response 17: Given the novelty of the first demonstration of fully integrated, high-performance optical clocks at sea, we feel the current title is appropriate and need not specify atomic/molecular species.

Line 18 “Operating high-performance clocks at sea has been historically challenging”

 “historically challenging” here might reference a bit too far back in time ( Harrison ?). For mechanical and even quartz clocks a ship might indeed be an especially challenging environment due to its motion, but does this still hold true for, e.g., atomic cesium or rubidium clocks? It would be better to instead present (although not in the abstract) a comparative overview of the challenges for mobile clocks in various environments (surface ship, submarine, spacecraft, fixed wing aircraft, helicopter, automotive, ...)

Author Response 18: Added the following text to Supplemental Material that discusses the Operation of Clocks on Mobile Platforms:

While optical clocks have been demonstrated outside of the laboratory [47], their use on moving platforms has been limited. While not as accurate as leading laboratory clocks, vapor cell clocks such as iodine offer practical advantages and enable state-of-the-art performance in mobile applications. Vapor spectroscopy operates continuously in an iodine clock, with high bandwidth (> 10 kHz), low orientation shift ($\sim 10^{-14}/g$), and without a local oscillator (LO) flywheel in the system. Continuous operation eliminates the influence of noise aliasing through the Dick effect [48], which can put stringent requirements on the LO used in pulsed atomic clocks, especially under orientation changes and acceleration. Pulsed systems using conventional quartz LOs ($\sim 10^{-10}/g$) or rigidly mounted optical cavities ($\sim 10^{-11}/g$ [49] and [50]) are difficult to field on mobile platforms without sacrificing performance by increasing the atom servo bandwidth. Moreover, orientation changes pose significant challenges for many spectroscopic approaches (e.g., free falling cold atoms, or ultra narrow spectroscopy of trapped atoms/ions that rely heavily on LO performance), resulting in practical barriers for use outside the laboratory. Compared to existing commercial clocks that do operate on mobile platforms (e.g., Cs beam, RAFS), iodine offers a significant performance advantage and reduction in environmental sensitivity (e.g., magnetic fields, temperature). With adequate servo bandwidth above the platform dynamics, it is expected that iodine clocks can be deployed on land, sea, air, and space platforms with similar performance to the results presented here.

Line 32 “first demonstrations of optical clocks [19].”

 should also mention Nevsky, 10.1016/S0030-4018(01)01190-7

Author Response 19: Reference added.

Line 44 “shown in Figure 1A”

 Figure 1A does not really show clock prototypes, but rather a clock chassis. Sentence could read instead: Initial clock prototypes were integrated into 35 L, 3U 19-inch rackmount chassis, shown in Figure 1A.

Author Response 20: Clarified language:

Initial clock prototypes were integrated into 35 L, 3U 19-inch rackmount chassis, shown in Figure 1A.

Line 52-53 “A third clock (VIPER) was built using a smaller iodine spectrometer and laser system to reduce physics package SWaP at relaxed performance goals.”

 How much was S W and P reduced? By what measures and what are the size, the weight, and the power consumption of the clocks? Later in the text, only the SWaP of a new development is mentioned. Further, later in the text it is written that VIPER was built before PICKLES and EPIC... i.e., the SWaP was increased and the performance improved...

Author Response 21: Added clarifying text below:

Each system consumes ~ 85 Watts (excluding the external power supply) and weighs ~ 26 kilograms.

AND

A third clock (VIPER) with a relaxed performance goal of $< 5 \times 10^{-13} / \sqrt{\tau}$ was built using a smaller iodine spectrometer and simplified laser system to reduce the physics package volume by 50% and power consumption by 5 W; the chassis volume was unchanged.

Line 55 “Results”

 From the measurements performed with the clocks, can you provide information on the frequency reproducibility, retrace of the clocks?

Author Response 22: Added a plot of the measured drift over time for PICKLES-EPIC at NIST, RIMPAC, and our California facility (Figure 8) and corresponding text to Supplemental Material:

Figure 8 shows the progression of the relative PICKLES-EPIC fractional frequency, measured via optical beatnote at NIST, at-sea for RIMPAC, and at our California headquarters. The trendline represents the linear fractional frequency drift of $\sim -2 \times 10^{-15}$ /day that was observed during the 34-day NIST measurement and illustrates several interesting features. First, the relative drift during the RIMPAC underway is indistinguishable from the NIST drift. Second, the data from these three measurements approximately fall along the trendline extrapolated from the NIST drift. This suggests that the observed long-term drift does not depend upon the on-off state of the clocks, consistent with the drift origin arising from long term changes in the vapor cell. An in-house measurement of the helium collisional shift along with calculations of the helium permeation time constant for vapor cells of our geometry suggest a fractional frequency drift of $\sim 1-2 \times 10^{-15}$ /day for 1-2 years after filling. Finally, consistency between the NIST extrapolation and the RIMPAC and California data illustrate the reproducibility and robustness of the integrated system. The time periods separating these measurements consisted of ground transport of the clocks between Boulder, CO and California and then two air-freight shipments between California and Pearl Harbor, HI. Additionally, dedicated retrace measurements of the VIPER clock at our California facility indicate frequency reproducibility $< 2 \times 10^{-14}$ following a 4-hour off state.

Line 58 “operated on an optical table without any special environmental measures”

 No temperature stabilization at NIST Labs? It is later stated that: The laboratory housing PICKLES and EPIC was stable to 100’s of mK throughout the measurement campaign. Standard temperature stabilization could be mentioned here.

Author Response 23: No additional measures were taken to isolate the clocks from the testing environment at NIST. Added clarifying language that the clocks experienced that native environment:

The clocks operated on an optical table without any additional measures to insulate them from the NIST laboratory environment, which is temperature stabilized.

Line 80-82 “the drift rates ...”

 The drift rate looks surprisingly linear for such a long duration. Are there any ideas about the cause of the linear drift? Comparison of the linear drift to the accuracy of another iodine clock is not plausible.

Author Response 24: See Author Response 22.

Line 86-88 “Without drift removal, the long-term clock performance is competitive with active hydrogen masers; drift removal puts the clock stability on-par with the highest performing masers.”

 This is statement should be reconsidered, the second part is definitely not true (see e.g., <https://www.vremya-ch.com/index.php/en/products-en/activehm-en/vch-2021-en/index.html>). In general, Masers still seem to be better in the especially application relevant range between 1E4 and 1E6 seconds.

Author Response 25: The original comment in the manuscript was referencing the NIST masers in Figure 2 against which the iodine clocks were measured. The two lowest drift masers in the NIST timescale (ST05 and ST22) exhibit fractional instabilities within 2× of the de-drifted iodine clocks at 1e6 seconds; the iodine clock outperforms five of the NIST masers at this timescale.

The linked Vremya maser indeed has exceptional stability, but the Allan deviation is de-drifted (per the details in the referenced EFTF/IFCS proceeding below the graph), and the 1-day drift at turn-on is 2e-15, which is comparable to PICKLES.

To avoid any confusion, we have added context to make explicitly clear that the comparison was against the NIST masers seen in Figure 2:

Without drift removal, the long-term clock performance is competitive with the NIST active hydrogen masers; drift removal puts the clock instability on-par with the highest performing masers in the NIST bank.

Line 96-97 “7 hour timescales”

 To find the peak deviation in the Allan deviation plot, where the averaging time is given in seconds, stating "20 ks (~7 h)" might help.

Author Response 26: Specified the pertinent time in seconds:

A broad feature with a peak deviation of $\sim 4 \times 10^{-15}$ is evident in the PICKLES Allan deviation at $\sim 20,000$ s (~7-hour) timescales.

Line 97-98 “shift of the hyperfine transition linewidth”

 A shift of the 'linewidth' does not make physical sense. Likely here RAM introduces a shift in the lockpoint, i.e. an offset from the center of the transition, rather than a shift to the transition ... Also, the term 'hyperfine transition' could be reconsidered.

Author Response 27: Amended the text to read:

The equivalent ~ 2 Hz optical frequency deviation corresponds to a ~ 2 ppm shift of the hyperfine transition line center.

Caption Figure 3 (A) “Clock stack-up for RIMPAC 2022. The server rack contained three independent optical clocks”

 Can you elaborate on common mode rejection of environmental effects with the three clocks assembled close together in the same container? Did you consider comparison to GPS?

Author Response 28: Added analysis/text placing an upper bound on potential environmental correlations due to acceleration, temperature, and magnetic fields. This text is split between the main body and the Supplemental Material:

All three clocks were co-located for the at-sea testing, therefore, there is potential for correlated environmental sensitivities due to ship dynamics, motion in earth’s magnetic field, and temperature and humidity variations inside the Conex. Standard reference clocks (e.g., cesium

beam clock or GPS-disciplined rubidium) were not available for comparison. However, simultaneous evaluation of three clocks raises the level of common mode rejection required to mask fluctuations common to the three systems, particularly given VIPER's differing spectrometer and laser system designs. Pairing the at-sea test data of three clocks with environmental testing on land provides confidence potential correlations are below the measured instability (See Supplemental Material).

AND

The acceleration sensitivity for VIPER was measured on land at $< 1 \times 10^{-14}/g$. For timescales relevant to ship dynamics (timescales $< 10^3$ s), scaling the measured acceleration in Figure 3E by this coefficient implies that acceleration-induced shifts are $\sim 100\times$ below the clock noise. Timescales from $10^3 - 10^5$ are more challenging to analyze and contain regions where temperature variation is strongest. Using independent environmental chamber testing on land, temperature coefficients were measured for PICKLES ($< 5 \times 10^{-15}/^\circ\text{C}$), EPIC ($2 \times 10^{-14}/^\circ\text{C}$), and VIPER ($8 \times 10^{-14}/^\circ\text{C}$). Given this, common mode temperature fluctuations between clocks are relatively small, and the correlation between the Conex temperature fluctuations and clock instabilities is consistent with the measured temperature coefficients. Finally, measured drift rates consistent with the NIST measurements suggest that the instabilities for $> 10^5$ s timescales accurately reflect the intrinsic behavior of the clocks. Notably, at the level that correlations could exist, cesium beam clocks are not precise enough to identify them on these timescales (e.g., Figure 1, Green trace).

With respect to magnetic fields, VIPER's sensitivity has been measured at $< 10^{-14}/\text{Gauss}$ despite having no shield. EPIC and PICKLES include shielding which reduces the impact of Earth's field by $> 10\times$. Given the scale of the shifts and differences in the clock hardware, correlated magnetic sensitivity during the underway is very unlikely at these performance levels. Finally, no humidity-correlated frequency instabilities were observed among the three clocks despite $\sim 15\%$ changes over the ~ 10 -minute AC period and 4-5% over day-night cycles throughout the underway.

Line 99 "uncontrolled"

 What is a controlled etalon? Maybe "spurious" or "parasitic"?

Author Response 29: Modified the text to read:

We suspect that the origin of this plateau in PICKLES is RAM coupling through a spurious etalon in the spectrometer.

Line 106-107 "In addition, the iodine-stabilized laser provides ..."

 Compared to other papers cavities only a factor of 1000?
<https://iopscience.iop.org/article/10.7567/APEX.7.022705/meta> ULE Cavity with drift rate of 2.16 ± 0.09 kHz/d at 657 nm. The cited paper reports on the drift rate between two lasers locked to two ULE cavities which might not be a good measurement of the drift rate of an individual cavity. Maybe also move this comparison to the section on the drift results where the drift is already compared to RAFS...?

Author Response 30: Moved text to paragraph discussing drift of RAFS. Amended typical drift range of ULE cavities to be in line with those cited by Hirata; added reference to Marmet et al., which Hirata cites as "typical" for state-of-the-art ULE:

Moreover, the iodine-stabilized laser provides $\sim 10,000$ - $100,000\times$ lower drift rate as compared to typical ultralow expansion (ULE) optical cavities [31, 32].

Line 125 “3°C”

 Please specify if this is peak-to-peak or amplitude.

Author Response 31: Modified the text to read:

Although the Conex was air-conditioned, the internal environment underwent ~ 2 - 3 °C peak-to-peak temperature and 4-5% relative humidity swings over a day-night cycle.

Line 173 “To our knowledge, these clocks are the highest performing sea-based clocks to date.”

 Very carefully worded (“To our knowledge”), but with some probability there might actually be some higher performing sea-based clocks. Should check Russian and Chinese language publications (Masers ...), also follow up on <https://www.royalnavy.mod.uk/news-and-latest-activity/news/2022/march/11/220311-hms-pwls-quantum-technology>

Author Response 32: Our thorough literature search did not turn up any higher performing sea-based clock, including masers. We use standard verbiage “To our knowledge” because we cannot prove that a counterexample does not exist.

The MINAC, which the referee referenced, was aboard the ship for the RIMPAC demo. MINAC is a conventional, vapor cell microwave clock (<https://tinyurl.com/2tuckxue>) with an instability of $3E-11$ at 1 s, 400-600 \times higher than our iodine clocks.

Line 215 “oven”

 Is "oven" a good word here? Maybe a (thermo-electric) cooler?

Author Response 33: Modified text to read:

This is accomplished with a thermo-electric cooler that surrounds the cold finger and has been able to maintain out-of-loop fluctuations < 1 mK for time periods exceeding 10 days.

Figure 6

 There seems to be a distinct pattern in the frequency of the VIPER clock. Is there an explanation for this behavior? Is it correlated to maneuvers of the ship?

Author Response 34: The pattern noted corresponds to the air conditioning cycle in the Conex container. Modified text to read:

The PICKLES-EPIC data exhibits a temperature-driven instability in the $10^3 - 10^5$ s range due to insufficient air conditioner (AC) capacity during the day.

AND

VIPER exhibits a short-term instability of $1.3 \times 10^{-13} / \sqrt{\tau}$ as well as a more prominent diurnal temperature instability that peaks at 4×10^{-14} at $\sim 40,000$ s (corresponding to ~ 1 -day periodic instability).

References [37] and [38]

 could only find abstracts which do not provided details to the statements made!

Author Response 35: These statements were made in our in-person presentations and do not appear in the abstracts. As such, we have removed them as references.

Referee #4(Remarks to the Author):

Authors: We are pleased that the reviewer recognized the state-of-the-art combination of integration, robustness, and long-term performance of our clock. Their detailed feedback has helped us to strengthen the manuscript. Care has been taken to address their feedback while staying within the word count guidelines. Where fixes were suggested, we have addressed the referees' comments. Changes are identified in-line with the referee comments below and redlined in the revised .doc file.

We maintain that these optical clocks mark a significant advancement for quantum sensors owing to their combination of high performance, compact footprint, and environmental robustness. Their real-world operability can have broad societal and scientific impact for navigation, telecommunications, and astronomy. We reiterate that the impact of our work is technological rather than fundamental. Similar work has been featured in Nature [1], where device performance is not laboratory state-of-the-art, but actually fielding the instrument can have transformative utility, including for fundamental science.

[1] Nature 602, 590–594 (2022)

In this manuscript, the authors present a compact and robust implementation of an optical iodine clock. Iodine spectrometer, frequency comb and associated control electronics are combined in one single package, representing a near-product implementation of a frequency standard. They highlight their setups and show the measured performance, once in the lab at NIST and once during operation on a ship. The shown frequency instabilities of the iodine clock are comparable to those of an active hydrogen maser.

In my opinion, the presented implementation is very impressive regarding the obtained frequency stability in combination with compactness (35 l of volume) and robustness. Operation on a ship demonstrated the functionality under harsh environmental conditions (with respect to temperature variations and accelerations). The resulting frequency instabilities are comparable to the current state of the art in iodine references where – to my knowledge – long-term stabilities of up to 1.000.000 s have not yet been reported.

The manuscript is very well written and publication of the results is very welcome. It is recognized, that such a compact and easy-to-handle frequency standard will improve current applications of frequency standards and also open up new applications as stated in the manuscript. However, originality and relevance are not seen high enough for publication in Nature. Within Nature Portfolio Nature Communications, Light: Science and Applications, Communications Physics or Scientific Reports are considered more appropriate journals. The techniques used (iodine modulation transfer spectroscopy and Er fiber frequency comb) are well known and no novel techniques or methods are presented for the compact and robust implementation.

Going to more detail, I would like to add the following points:

1. The shown Allan deviation plots include curves with de-drifted data. What is the justification for this linear detrend? Without linear detrend of the data, the exclusiveness of the long-term stability is very much reduced.

Author Response 1: The linear drift is attributed to helium permeation. Added the following text to the Supplemental Material:

An in-house measurement of the helium collisional shift along with calculations of the helium permeation time constant for vapor cells of our geometry suggest a fractional frequency drift of $\sim 1-2 \times 10^{-15}$ /day for 1-2 years after filling.

Added a plot/analysis of the measured drift over time for PICKLES-EPIC at NIST, RIMPAC, and our California facility (Figure 8) and corresponding text to Supplemental Material:

Figure 8 shows the progression of the relative PICKLES-EPIC fractional frequency, measured via optical beatnote at NIST, at-sea for RIMPAC, and at our California headquarters. The trendline represents the linear fractional frequency drift of $\sim -2 \times 10^{-15}$ /day that was observed during the 34-day NIST measurement and illustrates several interesting features. First, the relative drift during the RIMPAC underway is indistinguishable from the NIST drift. Second, the data from these three measurements approximately fall along the trendline extrapolated from the NIST drift. This suggests that the observed long-term drift does not depend upon the on-off state of the clocks, consistent with the drift origin arising from long term changes in the vapor cell. An in-house measurement of the helium collisional shift along with calculations of the helium permeation time constant for vapor cells of our geometry suggest a fractional frequency drift of $\sim 1-2 \times 10^{-15}$ /day for 1-2 years after filling. Finally, consistency between the NIST extrapolation and the RIMPAC and California data illustrate the reproducibility and robustness of the integrated system. The time periods separating these measurements consisted of ground transport of the clocks between Boulder, CO and California and then two air-freight shipments between California and Pearl Harbor, HI. Additionally, dedicated retrace measurements of the VIPER clock at our California facility indicate frequency reproducibility $< 2 \times 10^{-14}$ following a 4-hour off state.

We note that without drift removal, the long-term clock performance is still competitive with the NIST active hydrogen masers; drift removal puts the clock stability on-par with the highest performing masers in the NIST bank.

2. Application of the iodine clock on GNSS satellites is stated but suitability and relevant delta steps for space compatibility of the implementation are not given.

Author Response 2: Added text to the Supplemental Material:

Given current performance and SWaP, these clocks can already find near-term use in GNSS ground stations. Future space operation will require an order-of-magnitude reduction in SWaP, and radiation tolerance. We are actively developing a 5 L clock based on the same core physics package. These clocks share much of the iodine cell and laser componentry used in past [51] [23] and future space missions [12], thus presenting no fundamental barriers to space operation.

3. The manuscript lacks technical details of their implementation that are of interest to the corresponding community. These include, for example, the length of the gas cell, the interaction pathlength within the cell, the beam diameter and the intensities of pump and probe beams.

Author Response 3: We have added several technical details to the manuscript regarding the iodine spectrometer and spectroscopy scheme. Examples include:

Light is sampled on the bench before the cell for implementing RAM stabilization.

Operation in dynamic environments highlights the robust, high-bandwidth clock readout (> 10 kHz control bandwidth) enabled by a vapor cell.

The PICKLES and EPIC spectrometers utilize a traditional vapor cell that includes a cold finger, which is temperature stabilized at a setpoint of ~ 5 °C to maintain iodine pressure.

The VIPER spectrometer employs a starved iodine cell, which carries a fixed quantity of gaseous iodine corresponding to a fill temperature of ~ 0 °C (~ 4 Pascals) [39] [40].

An acousto-optic modulator (AOM) is used to frequency offset the pump beam from the probe beam by ~ 200 MHz to avoid a coherent background signal due to spurious reflections [42].

The modulation frequency and frequency deviation are unique to each system, but typical values are ~ 300 kHz and ~ 1 MHz, respectively.

We also added a simplified spectrometer schematic (Figure 5) to Supplemental Material.

4. Since the clocks were operated in the same container (and in the same rack), they were subject to the same temperature variations. Were common-mode effects taken into account?

Author Response 4: Added analysis/text placing an upper bound on potential environmental correlations due to acceleration, temperature, and magnetic fields. This text is split between the main body and the Supplemental Material:

All three clocks were co-located for the at-sea testing, therefore, there is potential for correlated environmental sensitivities due to ship dynamics, motion in earth's magnetic field, and temperature and humidity variations inside the Conex. Standard reference clocks (e.g., cesium beam clock or GPS-disciplined rubidium) were not available for comparison. However, simultaneous evaluation of three clocks raises the level of common mode rejection required to mask fluctuations common to the three systems, particularly given VIPER's differing spectrometer and laser system designs. Pairing the at-sea test data of three clocks with environmental testing on land provides confidence potential correlations are below the measured instability (See Supplemental Material).

AND

The acceleration sensitivity for VIPER was measured on land at $< 1 \times 10^{-14}$ /g. For timescales relevant to ship dynamics (timescales $< 10^3$ s), scaling the measured acceleration in Figure 3E by this coefficient implies that acceleration-induced shifts are $\sim 100\times$ below the clock noise. Timescales from $10^3 - 10^5$ are more challenging to analyze and contain regions where temperature variation is strongest. Using independent environmental chamber testing on land, temperature coefficients were measured for PICKLES ($< 5 \times 10^{-15}$ /°C), EPIC (2×10^{-14} /°C), and VIPER (8×10^{-14} /°C). Given this, common mode temperature fluctuations between clocks are relatively small, and the correlation between the Conex temperature fluctuations and clock instabilities is consistent with the measured temperature coefficients. Finally, measured drift rates consistent with the NIST measurements suggest that the instabilities for $> 10^5$ s timescales accurately reflect the intrinsic behavior of the clocks. Notably, at the level that correlations could exist, cesium beam clocks are not precise enough to identify them on these timescales (e.g., Figure 1, Green trace).

With respect to magnetic fields, VIPER's sensitivity has been measured at $< 10^{-14}$ /Gauss despite having no shield. EPIC and PICKLES include shielding which reduces the impact of Earth's field by $> 10\times$. Given the scale of the shifts and differences in the clock hardware, correlated magnetic sensitivity during the underway is very unlikely at these performance levels. Finally, no humidity-correlated frequency instabilities were observed among the three clocks despite $\sim 15\%$ changes over the ~ 10 -minute AC period and 4-5% over day-night cycles throughout the underway.

Figure 1C shows the Allan deviation of the iodine clock at NIST and at sea. What is the underlying measurement (beat measurement between PICKLES and EPIC, three-cornered hat analysis)?

Author Response 5: Expanded the sentence into a new paragraph for clarity:

Summary data for PICKLES, the highest performing clock at NIST and at sea is shown in Figure 1C and D. Single clock performance at sea is comprised of the decorrelated instability for $\tau < 200$ s (Figure 3D – blue trace) and the PICKLES-EPIC data for longer periods (Figure 3D – black trace). The PICKLES-EPIC data is normalized by $1/\sqrt{2}$ as an upper bound for PICKLES assuming equal contributions. Notably, the performance of PICKLES is largely unchanged at sea.

5. It is stated, that the VIPER clock has smaller SWaP budgets and lower performance goals but no details on the different implementation is given. This would also be of interest with respect to the Allan deviations shown in Figure 3D.

Author Response 6. Added clarifying text:

A third clock (VIPER) with a relaxed performance goal of $< 5\times 10^{-13}/\sqrt{\tau}$ was built using a smaller iodine spectrometer and simplified laser system to reduce the physics package volume by 50% and power consumption by 5 W; the chassis volume was unchanged.

AND

Each system consumes ~ 85 Watts (excluding the external power supply) and weighs ~ 26 kilograms.

6. While the temperature stability for the PICKLES and EPIC iodine cells' cold fingers are given, this information lacks for the VIPER starved cell. (line 220)

Author Response 7: Added clarifying text:

Out-of-loop temperature fluctuations were maintained to < 10 mK throughout RIMPAC.

7. What is the temperature stability of the optical bench provided by the thermal shielding and at what absolute temperature is it operated? (line 225)

Author Response 8: Added text:

The thermal shield maintains < 1 mK out-of-loop temperature fluctuations on the optical bench for time periods exceeding 10 days.

8. More details on the used 1064nm laser source would be interesting, including type of laser, output power and linewidth. (line 241)

Author Response 9: We prefer not to identify the specific laser used, but added clarifying language to Methods:

Several different 1064 nm lasers have been evaluated for the laser front-end with similar observed short-term instabilities for the clock.

Reviewer Reports on the First Revision:

Referees' comments:

Referee #2 (Remarks to the Author):

I believe that the authors have satisfactorily addressed the questions and comments provided in the first review.

Referee #3 (Remarks to the Author):

Review of revised nature Manuscript#: 2023-08-15170A

Corresponding Author: Jonathan Roslund

Title: Optical Clocks at Sea

In their revised manuscript the authors have addressed all the critical points raised in my previous review and added some very interesting and valuable information. I thus now consider the manuscript suitable for publication.

Concerning the additional information now given, I would like to address only one point specifically:

The authors state that they have achieved a long term (exceeding 10 days) temperature stability of < 1mK for both, the cold-finger and the optical bench. This is a non-trivial achievement by itself and likely one of the important ingredients for obtaining the excellent long-term frequency stability. Could the authors thus maybe provide more information on how this was achieved (and how it was verified by out-of-loop measurements)?